# Migraine Genetic Susceptibility Does Not Strongly Influence Migraine Characteristics and Outcomes in a Treated, Real-World, Community Cohort

**DOI:** 10.3390/jcm14020536

**Published:** 2025-01-16

**Authors:** Bruce A. Chase, Roberta Frigerio, Susan Rubin, Irene Semenov, Steven Meyers, Angela Mark, Thomas Freedom, Revital Marcus, Rima Dafer, Jun Wei, Siqun L. Zheng, Jianfeng Xu, Ashley J. Mulford, Alan R. Sanders, Anna Pham, Alexander Epshteyn, Demetrius Maraganore, Katerina Markopoulou

**Affiliations:** 1Department of Information Technology, Endeavor Health, Skokie, IL 60077, USA; 2Pritzker School of Medicine, Chicago, IL 60637, USA; 3Research Institute, Endeavor Health, Evanston, IL 60201, USA; 4Department of Neurology, Endeavor Health, Evanston, IL 60201, USA; 5University of Chicago Pritzker School of Medicine, Chicago, IL 60637, USA; 6Department of Neurological Sciences, Rush University Medical Center, Chicago, IL 60612, USA; 7Center for Individualized Medicine, Endeavor Health, Evanston, IL 60201, USA; 8Genomic Health Initiative, Endeavor Health, Evanston, IL 60201, USA; 9Department of Psychiatry and Behavioral Neuroscience, University of Chicago, Chicago, IL 60637, USA; 10Department of Neurology, Tulane University, New Orleans, LA 70118, USA

**Keywords:** migraine, polygenic risk score, real-world study, structured clinical documentation support tools, electronic health record (EHR) review

## Abstract

**Background/Objectives**: Migraine is a common neurological disorder with highly variable characteristics. While genome-wide association studies have identified genetic risk factors that implicate underlying pathways, the influence of genetic susceptibility on disease characteristics or treatment response is incompletely understood. We examined the relationships between a previously developed standardized integrative migraine polygenic genetic risk score (PRS) and migraine characteristics in a real-world, treated patient cohort. **Methods**: This retrospective cohort study used covariate-adjusted regression to comprehensively evaluate associations between the PRS and clinical characteristics in 1653 treated migraine cases with European ancestry at baseline and, in 800 cases, after one year. Cases were deeply phenotyped by neurologists during extensive interviews, using structured clinical documentation tools to record ~200 discrete data elements. **Results**: In treated patients, higher standardized PRS showed associations with two common migraine symptoms: photophobia (odds ratio [confidence interval]: 1.33 [1.13–1.56], *p* = 0.001) and stabbing pain (1.21 [1.08–1.36], *p* = 0.001]; both retained significance at Q = 0.05. Associations with phonophobia, nausea, emesis, and unilateral headache had similar effect sizes but did not survive correction for multiple tests. In this population, the PRS was not associated with other symptoms of migraine attacks, objective measures of migraine disability, frequency, severity, average duration, time-to-peak intensity of migraine attacks, chronification, emergency department visits, triptan responsiveness, or changes at follow-up. **Conclusions**: In treated patients, genetic risk was associated with common migraine symptoms but not with the severity of migraine characteristics or treatment outcomes. This suggests that in treated patients, other genetic and non-genetic factors influence migraine symptom severity and disease course more strongly than genetic susceptibility.

## 1. Introduction

Migraine is a prevalent neurologic disorder with highly variable presentation, symptom severity, and treatment response. Diagnosis of migraine is based on International Classification of Headache Disorders (ICHD-3) consensus criteria, derived by the International Headache Society [1]. Diagnostic criteria include moderate or severe pulsating pain, unilateral location, photophobia, phonophobia, nausea/vomiting, and a duration of 4–72 h (without treatment). Migraine is considered a complex genetic disorder with strong familial aggregation, reflecting a combination of genetic, behavioral, and environmental factors, each with small effect sizes. Although genetic variants are associated with the risk of developing migraine, how genetic susceptibility influences migraine characteristics is less thoroughly studied.

From both clinical and research perspectives, it would be useful to understand the strength of association between genetic risk factors for migraine and migraine-associated phenotypes, treatment responses, and comorbidities [2,3,4]. With notable exceptions such as familial hemiplegic migraine, where the inheritance of mutations in the *CACNA1A*, *ATTP1A2* or *SCN1A* genes leads to a phenotype that includes reversible motor weakness, migraine has not been associated with single genes. However, genome-wide association studies (GWAS) have identified at least 181 variants (single nucleotide polymorphisms or SNPs) associated with migraine susceptibility [5,6,7]. Most are characterized in populations of European descent so may not translate to risk in individuals with different ancestry [8,9]. A polygenic risk score (PRS) captures the combined risk at many potentially non-independent loci with effect sizes that may not reach genome-wide significance [10]. Though high-risk scores may not automatically predict disease characteristics, severity, or treatment response [11,12], they may provide information with clinical or personalized utility [2,13,14]. For example, determining genetic risk for triptan responsiveness could lead to individualized migraine treatment [2,15,16].

In a study of migraine families, PRS correlated with number and severity of migraine symptoms (number of attacks, duration, headache characteristics, nausea, vomiting, photophobia, and phonophobia) [17]. An outstanding question is whether migraine genetic risk scores are also associated with migraine symptoms and disease course in unrelated, treated patients.

In this cohort study, we tested the hypotheses that migraine genetic susceptibility is associated with clinical characteristics or outcomes in treated patients. We evaluated PRS associations in a deeply phenotyped patient cohort (N = 1653 cases) drawn from a real-world, treated, community-based practice setting, using an integrated PRS, PGS004799 [18], designed to leverage the power of several PRSs.

## 2. Materials and Methods

### 2.1. Study Patients, Migraine Diagnostic Criteria, and Data Elements

In this retrospective clinical and genetic study, we screened 2262 patients enrolled in *The DodoNA Project: DNA Predictions to Improve Neurological Health* (DodoNA), who were followed using a structured clinical documentation (SCDS) toolkit for headache disorders that was embedded in the electronic health record (EHR) [19]. DodoNA investigates the contribution of genetic risk to the progression and outcomes of ten neurological diseases, including migraine. All patients included in the study were at least 18 years old, were residents of Lake or Cook counties, Illinois, USA, and provided written informed consent for genotyping and EHR review. Studies were approved by the NorthShore University HealthSystem Institutional Review Board (EH10-139, 26 April 2011). Data were collected from study approval through June 2023.

Migraine was diagnosed by a neurologist using International Headache Classification 3rd ed (ICHD-3) criteria. Patients were deeply phenotyped through extensive clinical interviews with their treating neurologist, who comprehensively evaluated clinical characteristics using the SCDS toolkit and recorded more than 200 discrete data elements (see Appendix A). Evaluations were conducted at baseline (initial visit) and at annual follow-ups for up to ten years. We present analyses of baseline and first annual follow-up visits.

Patients were excluded if they had a non-migraine headache disorder, if they had non-European (EUR) ancestry, if they were not genotyped, if missing risk-variant genotypes precluded PRS calculation, or if they had insufficient data (e.g., minimal data on objective measures of migraine disability, symptoms, or characteristics (N = 609)). A cohort study evaluated risk-score associations with clinical characteristics at baseline in the DodoNA migraine cohort (N = 1653) and, excluding patients without first-annual follow up data, after [median (range)] 1.1 (0.4–2.7) yrs. (N = 800). A flow diagram of patient enrollment is presented in Figure 1.

### 2.2. Migraine Polygenic Risk Score

Genotyping, quality control measures, imputation, and ancestry determination were as described previously [20,21]. As described more fully in [21], we calculated the PGS004799 (hereafter referred to as PRS), which was developed using the PRSmixPlus pipeline that integrates multiple PRSs for a target population and leads to improved accuracy [18,22], using 2,938,299 SNPs that had a minor allele frequency > 0.005, *r*^2^ > 0.8, and that were also present in the Phase 3 1000-Genome reference panel.

### 2.3. Statistical Analysis of Associations Between the PRS and Migraine Characteristics

We used covariate-adjusted regression analyses to evaluate associations between the PRS and assessments at initial and one-year follow-up visits. Our approach was to fit models for a migraine characteristic or assessment using the PRS and a set of covariates that were reasonably expected to explain some of the variance associated with that characteristic or assessment, so that we could determine whether the PRS was a significant predictor in this context and whether inclusion of the PRS improved model fit or the variance explained by the model. Assessments included test scores on five questionnaires, including Migraine Disability Assessment [MIDAS, MIDAS-A (log-transformed to adjust distribution) and MIDAS-B], Migraine Specific Quality of Life Questionnaire (MSQ), Center for Epidemiological Studies Depression Scale (CES-D), Generalized Anxiety Disorder-7 (GAD7), and Insomnia Severity Index (ISI); categorical evaluation of frequency, average migraine duration, time-to-peak intensity, and severity; and migraine subtypes, characteristics, and symptoms obtained via the SCDS toolkit (Appendix A). We did not record when migraine characteristics and migraine-associated symptoms occurred within a migraine attack so were unable to evaluate PRS associations with prodromal, headache, or postdromal phases. We did not evaluate associations with migraine symptoms or characteristics that were present in <5% of patients (exploding, imploding headache; face-flushing, red-eye, ptosis, or ringing ears; dysarthria, motor, or aphasia types of aura) and assessed associations with unilateral and bilateral headache, but not with specific regions (e.g., occipital or temporal regions). Analyses were adjusted for sex, age at initial visit, disease duration, body mass index (BMI), presence of aura (except in analyses of migraine subtypes), use of abortive and preventive medications, number of years of education, and comorbidities seen in at least one percent of patients. Table 1 and Figure 2, Figure 3 and Figure 4 describe the distribution of the variables and covariates in this patient population.

Power analyses used a baseline N = 1500 and a follow-up N = 700, assuming that not all patients would have data for all covariates retained in a model, so the number of patients included in a regression model would be less than the 1653 patients assessed at baseline or the 800 assessed at follow-up. Power analyses of multiple linear regressions using *power rsquared* in *Stata BE 18.0* found the minimum detectable value for the *R*^2^ of a model with 4–10 retained covariates with N = 1500, alpha = 0.05, and power = 0.80 to be 0.0079–0.0108, which corresponded to effect sizes (δ) of 0.0080–0.0109. For N = 700, the minimum detectable *R*^2^ was 0.0169–0.0230, corresponding to effect sizes of 0.0172–0.0235. Multiple linear regression tests using between 4 and 10 coefficients with an *R*^2^ from 0.05 to 0.30 and alpha = 0.05 all had power = 1 for N = 1500 and power ≥ 0.996 for N = 700. Logistic regression power analyses used *powerlog* in *Stata BE 18.0* and the observed squared multiple correlation between the standardized PRS and all of the evaluated covariates. These identified power > 0.80 for N = 1500 over a range from 0.05 to 0.95 for the probability of the response variable equaling 1 at standardized PRS = 0 (mean) and if that probability increased by ≥ 0.0113 at standardized PRS = 1 (mean + 1 standard deviation).

Regression models [*measure ~ PRS + covariates + ten genetic principal components (PCs)*] were fit using Akaike information criterion (AIC) backwards selection. In this approach, non-significant covariates (i.e., *p* ≥ 0.05) can be retained in a model if they produce an AIC closer to zero. Continuous covariates were standardized to facilitate comparison of effect sizes. The type of regression model was selected based on the distribution of the dependent variable (Figure 3 and Figure 4). If a model using baseline data was significant, a sensitivity analysis evaluated whether the dependent variable differed between the tenth (highest) and first (lowest) deciles of PRS scores (using the first decile as the reference). Non-significant models, including non-significant models in sensitivity analyses, and models failing a link test, a Ramsey Regression Equation Specification Error Test (RESET), or goodness-of-fit tests described below are not reported.

(i)Linear regression models evaluated associations with ln(MIDAS-A) and MSQ scores. These produced close to normally distributed residuals.(ii)Generalized negative binomial regression models were used to evaluate associations with count variables showing non-normal distributions: ISI, MIDAS, CES-D, and GAD-7. For these, a Pearson or deviance goodness-of-fit test first demonstrated that a Poisson model was inappropriate: a negative binomial mean-dispersion model validated that the count variable showed overdispersion relative to a Poisson model, and overdispersion was modeled selecting from the covariates. This regression approach models the log of the expected count of the over- or under-dispersed dependent variable as a function of the predictor variables. Hence, the coefficient of a predictor variable specifies how the log of the expected count changes when that predictor variable changes by one unit, while other predictor variables are held constant.(iii)Logistic regression models were used to evaluate associations with binomial outcomes: symptoms and characteristics of migraine attacks, migraine subtypes [migraine without aura (MoA), migraine with aura (MA), migraine with and without aura (MwwA), probable migraine], the presence of aura and types of aura, emergency department visits, hospitalization, chronification of migraine, and triptan responsiveness. Models for triptan responsiveness were adjusted for the number of triptans tried (Figure 3G). Models failing a Hosmer-Lemeshow goodness-of-fit test are not shown.(iv)Ordinal logistic regression (ordered proportional odds) models were used to evaluate associations with dependent variables having natural ordering: MIDAS-B and symptom severity (both ranked 0-low to 10-high), frequency (<1/month = 1, 1–3/month = 2, 1/wk = 3, 2–3/wk = 4, >3/wk = 5, daily = 6, constant = 7), average duration (<5 min = 1, 5–15 min = 2, 15–60 min = 3, 1–4 h = 4, 4–24 h = 5, 1–3 days = 6, >3 days = 7), time-to-peak intensity (<5 min = 1, 5–15 min = 2, 15–60 min = 3, 1–4 h = 4, >4 h = 5), and average aura duration (<5 min = 1, 5–60 min = 2, 1–3 h = 3, 3–24 h = 4, >24 h = 5). For these, odds ratios are reported for the first equation (level one) only. Models failing a Lipsitz goodness-of-fit test are not reported.

Prior to modeling PRS associations with differences at follow-up, we used a Wilcoxon matched-pairs signed-rank test to evaluate whether baseline and follow-up assessments differed. Time-to-peak intensity, CES-D score, GAD-7 score, and average duration of aura did not differ at follow-up (see Section 3), so we did not evaluate PRS associations for these measures at follow-up. To evaluate differences at follow-up for ISI scores, ln(MIDAS-A), frequency, severity, MIDAS-B scores, average duration of migraine attack, MSQ scores and MIDAS scores, we used the same regression approach used to model baseline associations, and we included the baseline value as a covariate. No models for MIDAS scores at follow-up passed a link test, and no models for average duration passed a Lipsitz test, so no associations with these measures at follow-up are reported.

We report data for the best fitting model: the number of patients retained in the model, i.e., those with data for the measure and the retained covariates, those with history for a binomial outcome, covariates retained and their coefficients or odds ratios, 95% confidence interval (CI) and unadjusted *p* value, and model *p*, AIC, and either *R*^2^ or McFadden’s pseudo-*R*^2^. For comparison, we also report model *p*, AIC, and pseudo-*R*^2^ for models with (or without) the PRS and its odds ratio or β and CI, and *p* value, if it was excluded (or retained) during model fitting. To assess the robustness of associations, we report associations using Benjamini-Hochberg false-discovery rate (FDR) adjusted Q = 0.05. Statistical analyses were performed using *Stata BE 18.0*.

## 3. Results

### 3.1. Study Participants

The studied patient cohort was 87.7% female, as is often seen in headache clinics and clinical trials [23,24,25], and comprised a treated patient population: at study enrollment, 97% of patients had disease duration > 1 yr, and 98.7% used an abortive (Table 1, Figure 2). All patients satisfied ICHD-3 diagnostic criteria for a primary headache disorder of migraine. Deep phenotyping at study enrollment and, in nearly 50% of the patients, at a one-year follow-up revealed that this cohort was diverse with respect to demographics (except for EUR ancestry), disease duration, comorbidities, migraine symptoms, objective measures of migraine-associated disability, and response to treatment regimens, and that ongoing treatment led to some patients reporting migraine duration < 4 h (Figure 2, Figure 3 and Figure 4). Correlations between different toolkit assessments capturing similar clinical information (e.g., Figure 2G) provided support for data robustness. The diversity of clinical phenotypes indicated that these patients were representative of a real world, community-based cohort under active medical management for migraine and thus provided a useful framework to assess whether migraine genetic risk is associated with migraine symptoms, characteristics, and outcomes in a treated patient population.

### 3.2. The PRS Was Associated with a Subset of Migraine Symptoms and Characteristics

When PRS associations with migraine symptoms and characteristics were evaluated using covariate-adjusted regression, a higher PRS was associated with common migraine characteristics and migraine-associated symptoms: photophobia, stabbing pain, phonophobia, nausea, emesis, and unilateral pain (Table 2; Figure 5). These associations had effect sizes (standardized-PRS ORs) ranging from 1.12 to 1.33; however, only the associations with photophobia and stabbing pain remained significant at Q = 0.05 (Appendix A). Sensitivity analyses assessed PRS associations comparing groups with tenth-decile (highest) and first-decile (lowest, reference) PRS scores. Membership in the group with tenth-decile PRS scores was associated with increased photophobia, stabbing pain, phonophobia, emesis, unilateral pain, osmophobia, and fatigue (Table 2). Hence, the results of sensitivity analyses were generally congruent with findings using the entire cohort and consistent with photophobia, stabbing pain, and phonophobia being present more often in the tenth-decile PRS group (Appendix A). In models where the PRS was retained (i.e., its inclusion improved model fit), its inclusion also increased McFadden’s pseudo-*R*^2^ by 6–20%.

For symptoms or migraine characteristics seen in more than a third of patients, the PRS was not associated with triptan responsiveness, aura (all types) or visual aura, pressure, or a throbbing or bilateral headache. While the group having first-decile (lowest) PRS scores had a lower percentage of triptan-responsive patients (decile 1: 30/91 (32.6%), deciles 2–10: 419/946 (44.2%); χ^2^(1) = 4.58, *p* = 0.032), this association was not significant when the number of triptans tried was included as a covariate in logistic regression. With the exception of stabbing pain, which was reported in 26% of patients, the PRS was not associated with symptoms reported in a third or less of patients: neck stiffness, allodynia, fatigue, visual changes, sensory aura, vise-like pain, dizziness, or dull aching pain. The association with stabbing pain was not due to an association between stabbing pain and a subtype of migraine [MoA, χ^2^(1) = 2.10, *p* = 0.147; MA, χ^2^(1) = 0.12, *p* = 0.728; MwwA, χ^2^(1) = 3.58, *p* = 0.059] or another primary headache type, such a tension-type headache that was present in 4.6% of patients (χ^2^(1) = 0.585, *p* = 0.444). Therefore, these results support the inference that the PRS was associated with a subset of typical migraine symptoms in this patient population.

### 3.3. The PRS Was Not Associated with Migraine Subtypes, Measures of Severity, Frequency, Disability, or Outcomes

The PRS was not associated with measures of migraine-attack frequency (ln(MIDAS-A), frequency measured as a categorical variable), disability (MSQ, MIDAS), migraine subtypes (MA, MoA, MwwA), outcomes (chronification, emergency-department visits; models for hospitalization were not significant), symptom severity or pain (MIDAS-B), time-to-peak intensity, or the average duration of migraine attack or aura (Table 2 and Table 3; Figure 5). Though the group with tenth-decile (highest) PRS scores had higher ln(MIDAS-A) scores (Appendix A) and inclusion of the PRS improved the fit of the models for ln(MIDAS-A) score in sensitivity analyses (Table 3), it was not significant in these models. Including the PRS (or the group with tenth-decile PRS scores) as a covariate also improved model fit (lower AIC) and mostly increased *R*^2^ or pseudo-*R*^2^ for frequency, triptan-responsiveness, symptom severity, and time-to-peak intensity. This may reflect characteristics of this patient population, sample size, or indirect relationships between these measures and symptoms that show stronger associations with the PRS.

### 3.4. The PRS Was Not Strongly Associated with Insomnia, Depression, or Anxiety

An association between lower PRS and higher ISI scores did not survive correction for multiple testing at an FDR of 0.05 (Table 3 and Appendix A). While ISI scores decreased at follow-up, the PRS was not a significant predictor and did not improve model fit or *R*^2^ (Table 4). Significant models could not be developed for CES-D or GAD-7 scores at baseline, and these scores did not change in the subset of patients with follow-up data, so models for follow-up were not developed. Therefore, the PRS did not appear to be associated with performance on validated scales assessing insomnia, depression, or anxiety.

### 3.5. The PRS Was Not Associated with Changes in Migraine Frequency, Severity/Pain, or Disability at Follow-Up

In the subset of patients with follow-up data, MIDAS scores, ln(MIDAS-A) and frequency, MIDAS-B scores and severity, and average duration of migraine attacks decreased, and MSQ scores increased. We were able to develop significant models fitting follow-up data for ISI, MSQ, ln(MIDAS-A), MIDAS-B, frequency, and severity. The PRS was not a significant predictor of change in these measures (Table 4, Figure 5). Inclusion of the PRS in models for ISI or MSQ scores slightly increased pseudo-*R*^2^ or *R*^2^, respectively, but had no effect on pseudo-*R*^2^ in models for frequency, MIDAS-B, or severity and did not improve model fit for any measure.

## 4. Discussion

Prior to this study, little was known about how genetic susceptibility to migraine is related to disease severity and outcomes in treated patients. We found that the PRS was associated with more common migraine characteristics and symptoms but not with objective measures of migraine disability at baseline or with changes at follow-up. More specifically, in this population, higher PRS was associated with photophobia and stabbing pain, but the PRS was not associated with other symptoms of migraine attacks, objective measures of migraine disability, frequency, severity, average duration, time-to-peak intensity, chronification, emergency department visits, triptan responsiveness, or, at follow-up, improvement in MIDAS or MSQ scores, migraine frequency, severity, or average duration. Given the high prevalence of migraine and the multiple factors that influence risk, our results suggest that in treated patients, non-genetic factors and genetic factors that are not necessarily related to migraine susceptibility more strongly influence migraine severity and disease course than genetic risk of susceptibility to migraine.

A latent class analysis failed to show an association of genetic variants with migraine symptoms [26]. Similarly, in other diseases, genetic risk for susceptibility and progression are at least partly distinct [13,27]. In our analyses, demographic variables were often significant covariates in associations of risk scores with measures of migraine disability, characteristics, and outcomes (Table 2, Table 3 and Table 4). In this regard, it is interesting that in our relatively well-educated study population (Figure 2D), years of education was retained in nearly all models, even when it was not a significant covariate. We recently reported that genetic susceptibility confers increased risk of migraine attacks throughout most of adult life, but not earlier chronification of migraine [21]. Thus, stratification by genetic susceptibility may be useful in prospective research to evaluate outcomes in the context of demographic and clinical attributes such as comorbidities [28], baseline characterization of disease severity [29], or specific treatment paradigms [4,16].

In this patient cohort, we found no association between triptan responsiveness and PGS004799, though responsiveness was associated with the number of triptans tried (OR[CI]: 2.5 [2.0–3.2], *p* < 0.001; Table 2). Previously, Kogelman et al. [2] found that a two-fold increase in migraine genetic risk, ascertained using a different PRS, was associated with a positive response to triptans. Since the number of patients we evaluated for triptan responsiveness (>1000) was sufficiently powered to detect the effect size observed in this earlier study, our results may reflect use of a different PRS (PGS004799); our inclusion/exclusion criteria, including the number of triptans tried as a covariate; or other attributes of our patient cohort. Nonetheless, treatment is expected to strongly influence migraine disability and symptoms. Since the PRS does not show robust associations with measures of migraine disability or with the improvement seen at follow-up in this treated patient population as assessed by changes in MSQ scores, ln(MIDAS-A), migraine frequency, MIDAS-B scores, and severity, our results suggest that treatment may ameliorate genetic risk.

In the clinical setting, risk score associations with migraine characteristics likely vary with treatment, environmental factors, and ancestry, though the latter is not well explored. The findings of our study suggest that clinicians should counsel patients with migraine that genetic susceptibility to migraine does not determine disease course. Addressing the genetic contribution to treatment outcome and disease course requires further research, however. More specifically, studying the trajectories of risk score–stratified patients who are deeply phenotyped at diagnosis, prior to the initiation of treatment, and obtaining detailed longitudinal follow-up will be important to clarify (i) whether migraine genetic susceptibility is associated with treatment refractoriness and (ii) whether risk scores capturing genetic variation in treatment-related pathways are associated with treatment response, and to dissect the relationship between genetic and environmental factors and treatment outcomes. Such prospective studies would have considerable potential to inform therapeutic management strategies.

### Limitations

Nearly 3 million SNPs were used in the calculation of the integrative PRS; however, including additional SNPs or using a different or more focused PRS might reveal different or stronger associations. To date, migraine PRSs do not have substantial predictive power [18,22,30].

This study was designed as a real-world, retrospective clinical and genetic investigation, and thus data collection was limited to data obtained from patients at a headache clinic in a community practice setting. Excluding individuals without evaluable data may increase the possibility of selection bias. However, as we expected missing data to be missing at random, this subset of patients was likely representative of the population. Use of a practice setting may also introduce bias by omitting persons with migraine who do not seek medical attention or whose unavailability at a particular follow-up reflects a potentially temporary symptom resolution.

Our study was hypothesis driven; however, our analyses were data driven and exploratory, as the variables used to specify and estimate our models were based on our data, and thus the resulting models may be specific to our population. Our results should therefore be interpreted in the context of our specific patient population and analysis strategy.

## Figures and Tables

**Figure 1 jcm-14-00536-f001:**
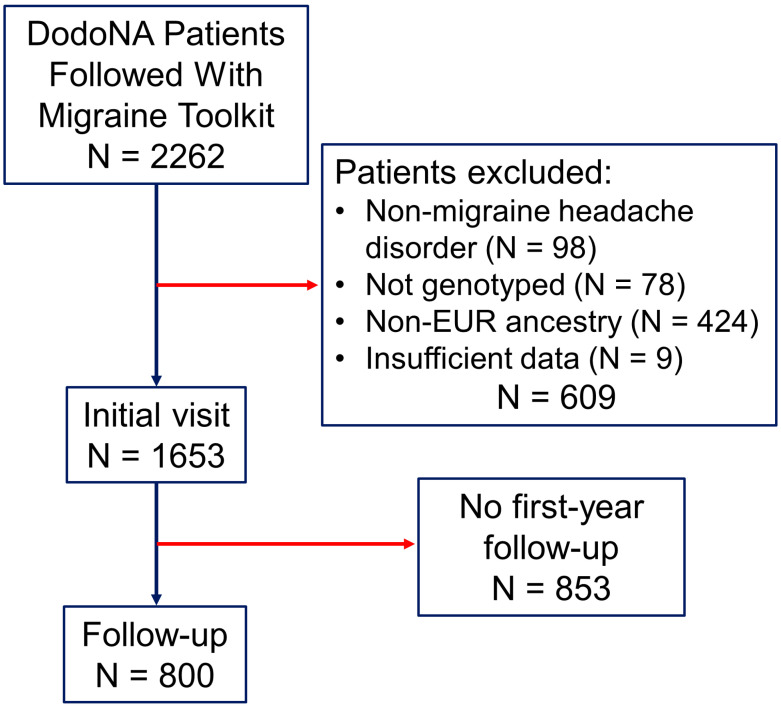
Study participants. Patients enrolled in the DodoNA project were recruited from communities in Lake and Cook counties, Illinois, USA. We screened 2262 patients followed by neurologists using a structured clinical documentation support toolkit developed for migraine and applying International Headache Classification 3rd ed. diagnostic criteria (Appendix A). Patients were included only if they had a primary headache disorder diagnosis of migraine and excluded if they had a non-migraine headache disorder, had insufficient clinical or genotype data, or had non-European (EUR) ancestry (since PGS004799 was developed and evaluated in individuals with this ancestry). Of the 1653 patients included at baseline, 800 also had data from a first annual follow-up (74 who lacked data at this follow-up had data at later follow-ups). Blue arrows: patients retained; red arrows: patients excluded or lost to follow-up.

**Figure 2 jcm-14-00536-f002:**
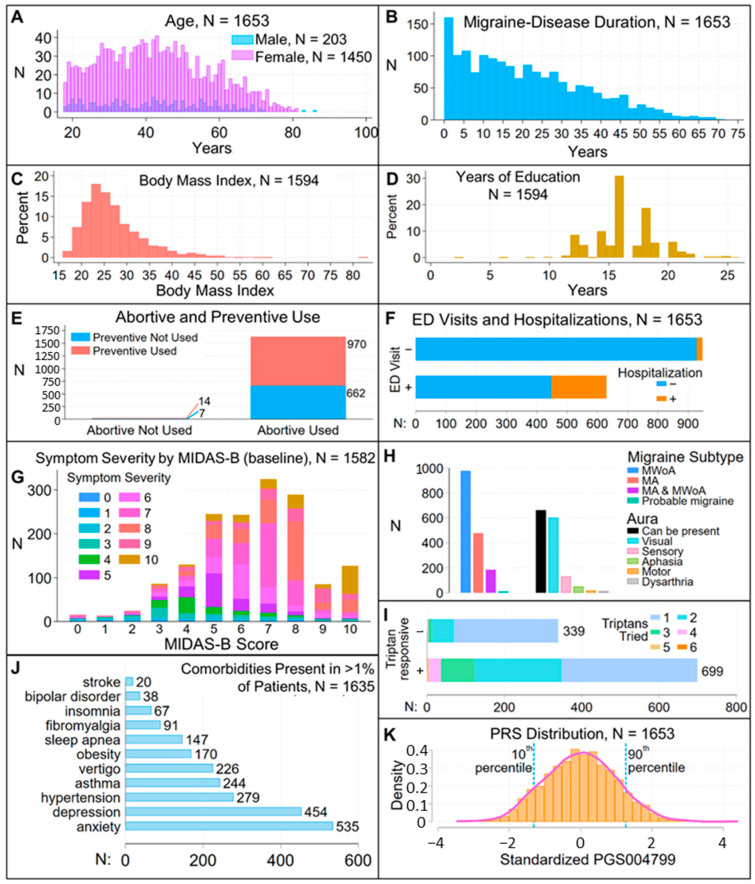
Distribution of demographic, clinical and genetic covariates. (**A**) Distribution of age, by sex, at baseline. (**B**) Duration of migraine disease at baseline. (**C**) Body mass index at baseline. (**D**) Years of education at baseline. (**E**) Use of abortives and preventives at baseline. (**F**) Emergency department (ED) visits and hospitalizations for migraine attacks through follow-up. (**G**) Symptom severity by MIDAS-B score, baseline data. These two symptomatic measures are moderately correlated (ρ = 0.51, *p* < 0.001). (**H**) Distribution of migraine subtypes and types of aura at baseline. MWoA: migraine without aura; MA: migraine with aura; MA and MWoA: migraine with and without aura. (**I**) History of response to triptans and the number of triptans tried, through follow-up. (**J**) Comorbidities present in more than one percent of patients at baseline. (**K**) Standardized PGS004799 distribution. The histogram is overlayed with a pink line showing a kernel density plot generated using an Epanechnikov kernel. Dashed blue lines identify 10th and 90th percentiles.

**Figure 3 jcm-14-00536-f003:**
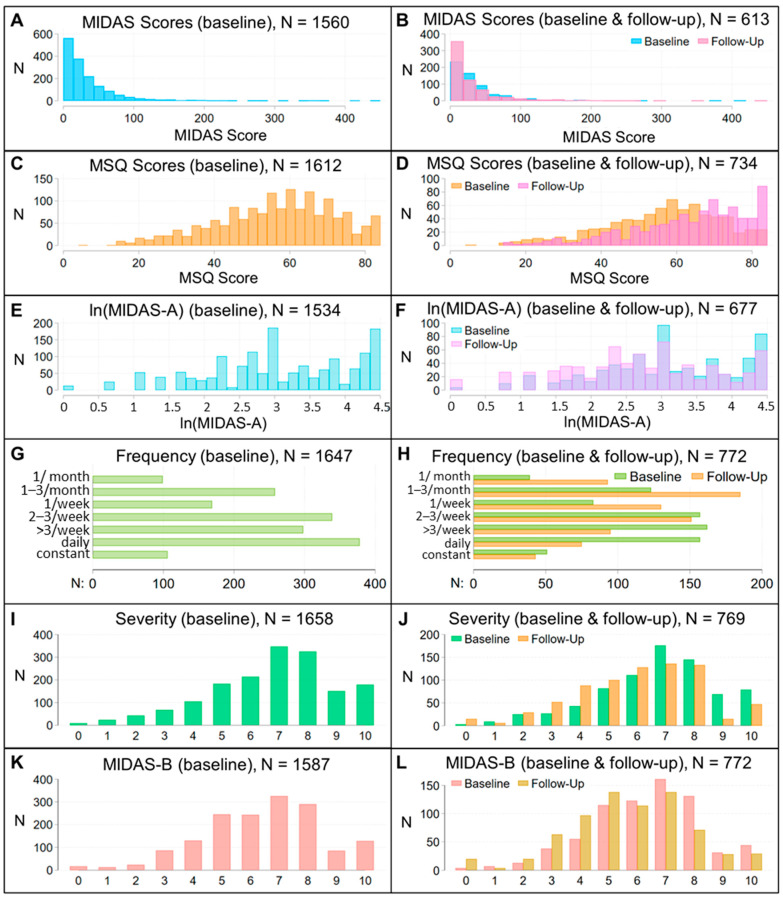
Measures of migraine disability, frequency, pain, and severity at baseline and in patients with baseline and follow-up data. Baseline (**A**) and follow-up (**B**) distribution of Migraine Disability Assessment Test (MIDAS) total scores. Baseline (**C**) and follow-up (**D**) distribution of Migraine Specific Quality of Life Questionnaire (MSQ) total scores. Baseline (**E**) and follow-up (**F**) distribution of ln(MIDAS-A) (headache days in three months). Baseline (**G**) and follow-up (**H**) distribution of frequency of migraine attacks. Baseline (**I**) and follow-up (**J**) distribution of symptom severity (0 low to 10-high). Baseline (**K**) and follow-up (**L**) distribution of MIDAS-B (pain) scores.

**Figure 4 jcm-14-00536-f004:**
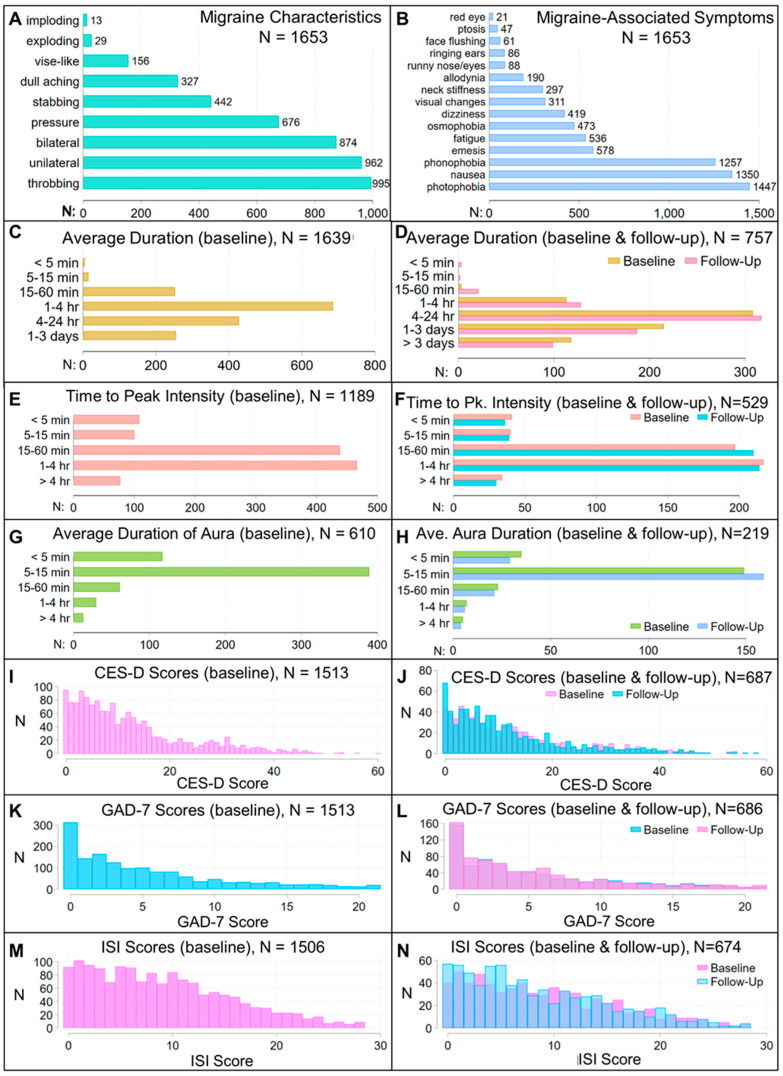
Distribution of migraine characteristics, migraine-associated symptoms, and scores on validated assessments of migraine comorbidities at baseline and in patients with baseline and follow-up data. (**A**) Migraine characteristics. (**B**) Migraine-associated symptoms. Baseline (**C**) and follow-up (**D**) distribution of the average duration of migraine attacks. In this treated patient population, duration <4 h reflects use of medication. Baseline (**E**) and follow-up (**F**) distribution of the time-to-peak migraine intensity. Baseline (**G**) and follow-up (**H**) distribution of the average duration of aura in patients who experience aura. Baseline (**I**) and follow-up (**J**) distribution of Center for Epidemiological Studies Depression Scale (CES-D) scores. Baseline (**K**) and follow-up (**L**) distribution of Generalized Anxiety Disorder 7-item (GAD-7) scores. Baseline (**M**) and follow-up (**N**) distribution of Insomnia Severity Index (ISI) scores.

**Figure 5 jcm-14-00536-f005:**
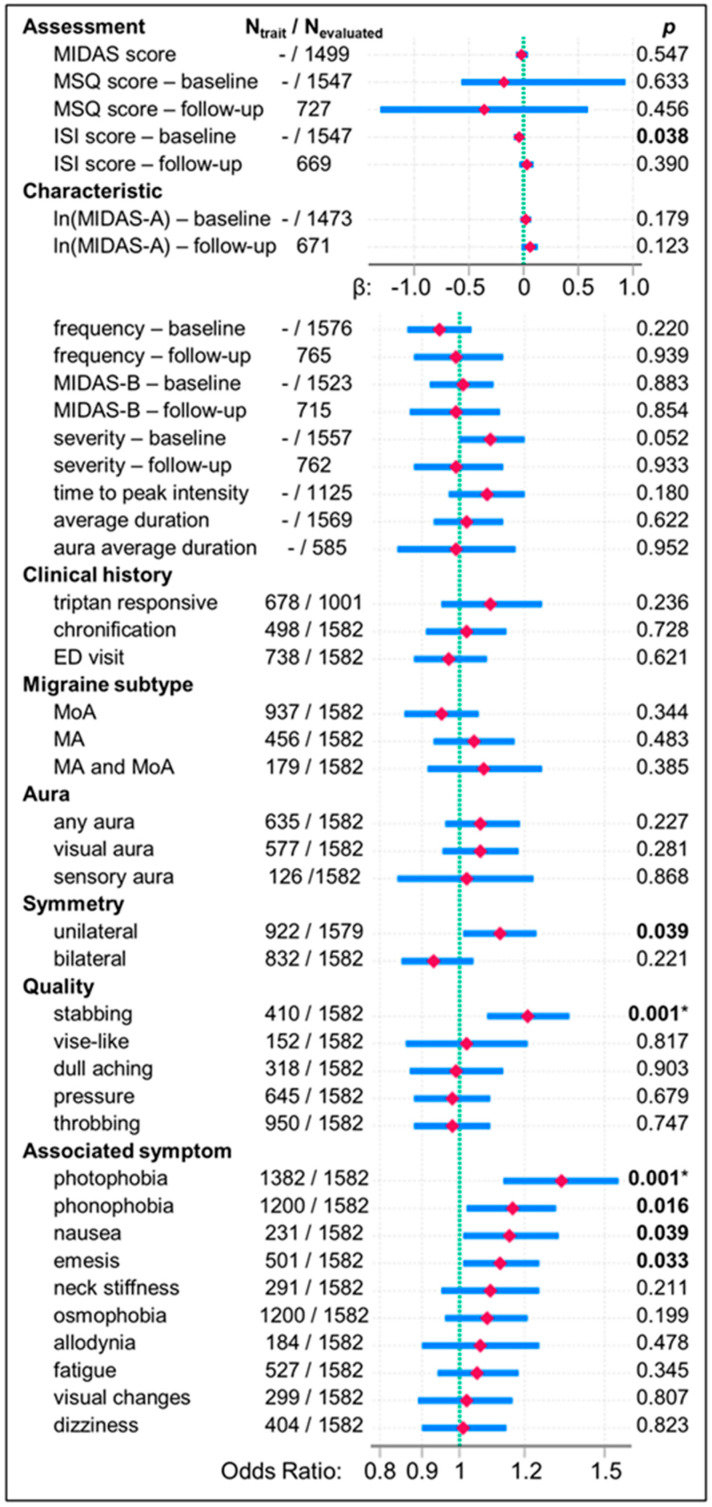
Summary of PRS associations. The panels summarize PRS associations identified using covariate-adjusted regression models using baseline and first annual follow-up data. A green dashed line marks β = 0 for linear and negative binomial regression models and an odds ratio = 1 for ordinal logistic and logistic regression models. Red diamonds mark the PRS estimate for β or odds ratio; blue lines show 95% confidence intervals. Values of *p* < 0.05 (uncorrected for multiple test tests) are in bold, and an asterisk indicates that the variable retained significance at Q = 0.05 (see Appendix A). The total number of patients included in each model is slightly less than the number of patients with data for a dependent variable (see Figure 2, Figure 3 and Figure 4) since regression models used only patients with complete data on all covariates retained in the model. For binomial dependent variables, N_trait_ is the number of patients with a positive medical history, characteristic, or symptom who were assessed in the model. N_evaluated_ is the number of patients evaluated in the model. Compare to details of models that are presented in Table 2, Table 3 and Table 4. Abbreviations: MIDAS, Migraine Disability Assessment Test; MSQ, Migraine-Specific Quality of Life Questionnaire; ISI, Insomnia Severity Index; ED, emergency department; MoA, migraine without aura; MA, migraine with aura; MA and MoA, migraine with and without aura.

**Table 1 jcm-14-00536-t001:** Variables analyzed in covariate-adjusted regression models.

Variable	N/Total (%)		Median (Range)	
Initial-visit age				
Male	203/1653 (12.3)		42.0 (18.1−86.6)	
Female	1450/1653 (87.7)		42.0 (17.1−81.2)	
Duration	1653		18.3 (0.1−71.6)	
Body mass index	1640		35.5 (15.8−83.2)	
Years of education	1594		16 (2−26)	
Migraine subtypes			-	
Migraine with aura	477			
Migraine without aura	980			
Migraine with and without aura	185			
Probable migraine	11			
Aura (any)	659/1653 (39.9)		-	
Visual	603			
Sensory	131			
Aphasia	49			
Motor	18			
Dysarthria	12			
Abortive use	1632/1653 (98.7)		-	
Preventive use	984/1653 (59.5)		-	
PGS004799	1653		0.8225 (0.0715−1.7674)	
Standardized PGS004799	1653		0.0079 (−3.4884−4.4076)	
Triptan responsive ^a^	699/1038 (67.3)		-	
Emergency department visit ^a^	761/1653 (46.0)		-	
Hospitalization for migraine ^a^	259/1653 (15.7)		-	
	*Baseline*	*Follow-up* ^b^	*Baseline*	*Follow-up*
MIDAS score	1560	613	21 (0−450)	16 (1–450)
MSQ score	1612	734	58 (4−84)	67 (14–84)
ln(MIDAS-A)	1535	677	3.04 (0–4.5)	2.71 (0–4.5)
CES-D score	1516	687	9 (0–60)	8 (0–58)
GAD-7 score	1513	686	4 (0–21)	3 (0–21)
ISI score	1506	674	8 (0–28)	7 (0–28)
Frequency	1647	773	4 (1–7)	4 (1–7)
<1/month	99	93		
1–3/month	258	186		
1/week	169	130		
2–3/week	339	151		
>3/week	298	95		
daily	278	75		
constant	106	43		
MIDAS-B	1587	743	7 (0–10)	6 (0–10)
0	17	21		
1	13	4		
2	24	21		
3	86	67		
4	139	98		
5	245	141		
6	243	116		
7	326	145		
8	290	72		
9	85	29		
10	128	29		
Severity	1648	772	7 (0–10)	6 (0–10)
0	9	16		
1	24	6		
2	43	29		
3	68	52		
4	105	88		
5	183	100		
6	214	128		
7	347	138		
8	325	133		
9	151	35		
10	179	47		
Average duration	1639	764	-	-
<5 min	0	3		
5–15 min	5	2		
15–60 min	15	22		
1–4 h	252	129		
4–24 h	685	319		
1–3 days	427	188		
>3 days	255	101		
Aura duration	610	303	-	-
<5 min	117	52		
5–15 min	389	207		
15–60 min	61	28		
1–4 h	30	8		
>4 h	13	8		
Time-to-peak intensity	1189	580	-	-
<5 min	108	38		
5–15 min	100	41		
15–60 min	438	228		
1–4 h	466	241		
>4 h	77	32		

^a^ Triptan responsiveness, hospitalization, and emergency department visits include baseline and follow-up data. ^b^ Follow-up N is larger than the N used to evaluate differences at follow-up, as the matched-pairs signed-rank test used only patients with data at both baseline and follow-up. Compare to Table 4 and Figure 3 and Figure 4.

**Table 2 jcm-14-00536-t002:** Association of migraine PRS with clinical history, migraine subtype, aura, symmetry, quality, and associated symptoms ^a^.

Symptom	Group	Symptom Present N/Model N	Covariate	Odds Ratio, 95% CI	*p*	Model		
						*p*	AIC	McFadden’s Pseudo *R*^2^
**A.** **Clinical history**								
History of chronic migraine	All	498/1582	Duration	1.30 (1.16, 1.46)	<**0.001**	<**0.001**	1799.56	0.1031
			Education years	0.84 (0.75, 0.94)	**0.003**			
			Female sex	1.81 (1.21, 2.72)	**0.004**			
			Preventive use	3.89 (2.99, 5.06)	<**0.001**			
			Depression	1.41 (0.98, 1.01)	**0.005**			
			+PRS	1.02 (0.91, 1.14)	0.728	<**0.001**	1801.44	0.1032
	PRS deciles 1 and 10	98/321	Duration	1.51 (1.17, 1.94)	**0.001**	<**0.001**	365.63	0.1503
			Education years	0.77 (0.59, 1.02)	0.071			
			Female sex	3.89 (1.29, 11.7)	**0.016**			
			Preventive use	3.88 (2.06, 7.33)	<**0.001**			
			+PRS decile 10	1.25 (0.72, 2.19)	0.429	<**0.001**	367.00	0.1519
Emergency department visit	All	738/1582	Education years	0.95 (0.86, 1.05)	0.300	**0.007**	2186.98	0.0073
			Abortive use	3.66 (1.21, 11.1)	**0.022**			
			Stroke	0.32 (0.11, 0.89)	**0.029**			
			Sleep apnea	1.70 (1.18, 2.44)	**0.004**			
			Obesity	0.76 (0.54, 1.07)	0.121			
			Hypertension	1.27 (0.97, 1.67)	0.086			
			+PRS	0.97 (0.88, 1.08)	0.621	**0.010**	2188.74	0.0152
	PRS deciles 1 and 10	151/319	Age	0.82 (0.65, 1.03)	0.091	**0.020**	445.03	0.0641
			BMI	1.12 (0.89, 1.41)	0.341			
			Education years	0.76 (0.61, 0.99)	**0.038**			
			Female sex	0.36 (0.18, 0.74)	**0.006**			
			Depression	1.56 (0.91, 2.68)	0.105			
			+PRS decile 10	1.23 (0.76, 1.99)	0.411	**0.024**	446.35	0.0656
Triptan responsive	All	678/1001	Duration	1.26 (1.08, 1.46)	**0.003**	<**0.001**	1171.66	0.0948
			Education years	1.14 (0.98, 1.31)	0.085			
			Triptans Tried	2.53 (2.00, 3.21)	<**0.001**			
			Aura	0.73 (0.54, 0.97)	**0.032**			
			Fibromyalgia	0.55 (0.32, 0.96)	**0.036**			
			+PRS	1.09 (0.95, 1.26)	0.236	<**0.001**	1172.24	0.0959
	PRS deciles 1 and 10	142/201	Duration	1.54 (1.07, 2.20)	**0.020**	<**0.001**	231.72	0.1874
			Education years	0.93 (0.64, 1.34)	0.692			
			PRS decile 10	1.93 (0.92, 3.06)	0.082			
			Triptans tried	2.33 (1.30, 4.17)	**0.004**			
			Fibromyalgia	0.28 (0.07, 1.10)	0.067			
			Asthma	2.10 (0.74, 6.00)	0.164			
			−PRS decile 10	–	–	<**0.001**	232.78	0.1784
**B.** **Migraine subtype**								
Migraine without aura (MoA)	All	937/1582	Education years	0.90 (0.81, 0.99)	**0.040**	<**0.001**	2129.30	0.0185
			Abortive use	3.19 (1.20, 8.46)	**0.020**			
			Preventive use	1.32 (1.07, 1.63)	**0.009**			
			vertigo	0.74 (0.55, 0.98)	**0.040**			
			+PRS	0.95 (0.86, 1.06)	0.344	<**0.001**	2130.40	0.0189
	PRS deciles 1 and 10	183/319	Education years	0.66 (0.51, 0.86)	**0.002**	**0.007**	436.15	0.0761
			Female sex	2.11 (1.05, 4.24)	**0.035**			
			Depression	0.51 (0.29, 0.90)	**0.021**			
			Fibromyalgia	0.30 (0.10, 0.86)	**0.025**			
			Bipolar disorder	8.67 (0.76, 99.6)	0.083			
			Insomnia	3.77 (0.88, 16.1)	0.074			
			+PRS decile 10	1.04 (0.63, 1.70)	0.891	0.011	438.13	0.0761
Migraine with aura (MA)	All	456/1582	Education years	1.04 (0.93, 1.17)	0.457	<**0.001**	1892.86	0.0207
			Female sex	0.71 (0.51, 0.99)	**0.041**			
			Abortive use	0.23 (0.09, 0.58)	**0.002**			
			Preventive use	0.75 (0.60, 0.94)	**0.014**			
			Vertigo	1.19 (0.97, 1.63)	0.284			
			+PRS	1.04 (0.93, 1.17)	0.483	<**0.001**	1894.37	0.0210
	PRS deciles 1 and 10	98/319	Education years	1.28 (0.99, 1.66)	0.063	**0.025**	397.46	0.0663
			Female sex	0.46 (0.23, 0.94)	**0.033**			
			Abortive use	0.13 (0.01, 1.39)	0.091			
			Fibromyalgia	1.92 (0.69, 5.37)	0.211			
			+PRS decile 10	1.18 (0.70, 2.00)	0.541	**0.033**	399.09	0.0672
Migraine with and without aura	All	179/1582	Education years	0.82 (0.67, 1.00)	0.053	**0.023**	1120.69	0.0236
			Body mass index	1.20 (1.03, 1.41)	**0.022**			
			Obesity	1.74 (0.87, 3.13)	0.064			
			Vertigo	1.37 (0.91, 2.08)	0.133			
			+PRS	1.07 (0.91, 1.26)	0.385	**0.028**	1121.94	0.0243
**C.** **Aura**								
Aura (any)	All	635/1582	Education years	1.12 (1.01, 1.24)	**0.031**	<**0.001**	2123.58	0.0176
			Abortive use	0.30 (0.11, 0.81)	**0.017**			
			Preventive use	0.78 (0.63, 0.96)	**0.017**			
			Vertigo	1.33 (0.99 (1.78)	0.055			
			+PRS	1.07 (0.96, 1.18)	0.227	<**0.001**	2124.12	0.0183
	PRS deciles 1 and 10	41/319	Education years	1.50 (1.16, 1.95)	**0.002**	**0.004**	435.15	0.083
			Female sex	0.50 (0.25, 1.02)	0.056			
			Abortive use	0.18 (0.02, 2.08)	0.172			
			Preventive use	0.66 (0.39, 1.10)	0.110			
			Depression	2.05 (1.15, 3. 66)	**0.015**			
			Fibromyalgia	4.05 (1.38, 11.8)	**0.011**			
			Bipolar disorder	0.11 (0.01, 1.29)	0.079			
			Insomnia	0.27 (0.05, 1.16)	0.078			
			+PRS decile 10	1.09 (0.66, 1.80)	0.744	**0.007**	437.05	0.0866
Visual aura	All	577/1582	Education years	1.12 (1.01, 1.25)	**0.034**	<**0.001**	2065.88	0.0202
			Abortive use	0.26 (0.10, 0.70)	**0.007**			
			Preventive use	0.81 (0.65, 0.99)	**0.047**			
			Depression	0.83 (0.65, 1.05)	0.124			
			Vertigo	1.32 (0.98, 1.78)	0.069			
			+PRS	1.06 (0.95, 1.18)	0.281	<**0.001**	2066.72	0.0208
	PRS deciles 1 and 10	125/319	Education years	1.51 (1.16, 1.96)	**0.002**	**0.002**	424.40	0.0955
			Female sex	0.58 (0.29, 1.19)	0.139			
			Depression	2.07 (1.16, 3.69)	**0.013**			
			Fibromyalgia	3.26 (1.13, 9.37)	**0.028**			
			Obesity	0.41 (0.19, 0.91)	**0.027**			
			Bipolar disorder	0.11 (0.01, 1.42)	0.090			
			Insomnia	0.33 (0.08, 1.40)	0.132			
			Hypertension	1.83 (0.98, 3.42)	0.059			
			+PRS decile 10	1.00 (0.60, 1.66)	0.986	**0.003**	426.40	0.0955
Sensory aura	All	126/1582	Education years	1.12 (0.93, 1.35)	0.224	**0.013**	880.41	0.0306
			Preventive use	0.51 (0.35, 0.74)	<**0.001**			
			Anxiety	1.50 (1.03, 2.19)	**0.035**			
			+PRS	1.02 (0.84, 1.23)	0.868	**0.020**	882.39	0.0306
**D.** **Symmetry**								
Unilateral	All	922/1579	Age	1.20 (1.07, 1.34)	**0.002**	<**0.001**	2128.78	0.0240
			Education years	1.09 (0.98, 1.21)	0.120			
			PRS	1.12 (1.01, 1.24)	**0.039**			
			Stroke	0.50 (0.20, 1.25)	0.137			
			Sleep apnea	0.75 (0.52, 1.08)	0.117			
			Obesity	0.72 (0.51, 1.01)	0.060			
			Hypertension	0.78 (0.58, 1.04)	0.095			
			−PRS	–	–	<**0.001**	2131.07	0.0220
	PRS deciles 1 and 10	183/319	Age	1.25 (0.99, 1.58)	0.066	**0.008**	437.12	0.0785
			Education years	1.13 (0.88, 1.43)	0.335			
			PRS decile 10	1.78 (1.08, 2.92)	0.023			
			Female sex	1.69 (0.83, 3.40)	0.146			
			Depression	1.56 (0.87, 2.82)	0.137			
			Anxiety	0.51 (0.30, 0.87)	**0.014**			
			Obesity	0.44 (0.22, 0.90)	**0.024**			
			−PRS decile 10	–	–	**0.024**	440.35	0.0665
Bilateral	All	832/1582	Education years	0.90 (0.81, 1.00)	**0.049**	**0.002**	2184.73	0.0165
			Aura	0.78 (0.64, 0.96)	**0.017**			
			Anxiety	1.19 (0.97, 1.48)	0.102			
			Stroke	2.07 (0.78, 5.47)	0.141			
			Obesity	1.36 (0.98, 1.90)	0.068			
			+PRS	0.93 (0.85, 1.04)	0.221	**0.002**	2185.24	0.0172
**E.** **Quality**								
Stabbing	All	410/1582	Age	0.86 (0.73, 1.00)	0.051	<**0.001**	1797.09	0.0283
			Duration	1.14 (0.98, 1.33)	0.097			
			Education years	1.05 (0.93, 1.18)	0.440			
			PRS	1.21 (1.08, 1.36)	**0.001**			
			Aura	1.22 (0.97, 1.54)	0.093			
			Abortive use	5.58 (0.73, 42.5)	0.097			
			Depression	1.32 (1.03, 1.70)	**0.028**			
			Vertigo	1.39 (1.01, 1.91)	**0.046**			
			−PRS	–	–	<**0.001**	1805.30	0.0227
	PRS deciles 1 and 10	92/319	Duration	1.20 (0.94, 1.54)	0.136	**0.017**	385.87	0.0687
			Education years	0.87 (0.67, 1.14)	0.322			
			PRS decile 10	2.32 (1.34, 4.03)	**0.003**			
			Fibromyalgia	2.51 (0.92, 6.87)	0.072			
			−PRS decile 10	–	–	0.153	393.15	0.0472
Vise-like	All	152/1582	Age	1.28 (1.08, 1.52)	**0.004**	**0.001**	995.87	0.0391
			BMI	0.86 (0.71, 1.05)	0.132			
			Education years	1.02 (0.85, 1.21)	0.815			
			Abortive use	0.37 (0.12, 1.18)	0.093			
			Preventive use	2.14 (1.44, 3.18)	<**0.001**			
			Anxiety	1.34 (0.95, 1.91)	0.098			
			PRS	1.02 (0.86, 1.21)	0.817	**0.002**	997.74	0.0393
	PRS deciles 1 and 10	36/319	Education years	0.91 (0.62, 1.33)	0.997	**0.010**	225.01	0.1238
			Preventive use	5.04 (1.68, 15.1)	**0.002**			
			Vertigo	2.07 (0.87, 4.91)	**0.031**			
			+PRS decile 10	1.70 (0.74, 3.90)	0.207	**0.009**	225.38	0.1311
Dull aching	All	318/1582	Duration	1.20 (1.06, 1.36)	**0.004**	<**0.001**	1544.68	0.0498
			BMI	0.70 (0.59, 0.83)	<**0.001**			
			Education years	1.13 (1.00, 1.29)	0.058			
			Aura	1.35 (1.04, 1.74)	**0.023**			
			Anxiety	1.58 (1.21, 2.05)	**0.001**			
			Obesity	2.20 (1.36, 3.57)	**0.001**			
			Vertigo	1.66 (1.19, 2.31)	**0.002**			
			+PRS	0.99 (0.87, 1.13)	0.903	<**0.001**	1546.66	0.0498
	PRS deciles 1 and 10	75/319	Age	1.35 (1.04, 1.77)	**0.024**	**0.006**	347.67	0.0928
			BMI	0.59 (0.41, 0.86)	**0.007**			
			Education years	1.05 (0.80, 1.39)	0.717			
			Aura	2.07 (1.18, 3.65)	**0.011**			
			Obesity	2.56 (0.93 7.06)	0.069			
			+PRS decile 10	1.00 (0.85, 1.18)	0.965	**0.009**	349.67	0.0928
Pressure	All	645/1582	BMI	1.24 (1.09, 1.40)	**0.001**	<**0.001**	2120.42	0.0273
			Education years	0.96 (0.87, 1.07)	0.463			
			Female sex	0.65 (0.48, 0.89)	**0.007**			
			Abortive use	2.11 (0.75, 5.93)	0.158			
			Preventive use	0.82 (0.67, 1.03)	0.090			
			Anxiety	0.83 (0.67, 1.04)	0.102			
			Fibromyalgia	1.63 (1.05, 2.55)	**0.030**			
			Sleep apnea	1.37 (0.95, 1.97)	0.096			
			Obesity	0.74 (0.50, 1.09)	0.128			
			+PRS	0.98 (0.88, 1.09)	0.679	<**0.001**	2122.24	0.0274
	PRS deciles 1 and 10	129/319	Education years	1.00 (0.79, 1.27)	0.997	**0.014**	432.93	0.0687
			BMI	1.59 (1.19, 2.13)	**0.002**			
			Sleep apnea	2.51 (1.09, 5.78)	**0.031**			
			Obesity	0.47 (0.19, 1.11)	0.085			
			Hypertension	0.56 (0.29, 1.07)	0.080			
			+PRS decile 10	0.95 (0.83, 1.10)	0.679	**0.018**	434.46	0.0698
Throbbing	All	950/1582	Age	0.74 (0.67, 0.83)	<**0.001**	<**0.001**	2101.73	0.0287
			Education years	1.03 (0.93, 1.14)	0.599			
			Aura	0.80 (0.65, 0.98)	**0.034**			
			Abortive use	2.37 (0.91, 6.15)	0.077			
			Preventive use	1.23 (1.00, 2.11)	0.053			
			Sleep apnea	1.46 (1.00, 2.11)	**0.048**			
			+PRS	0.98 (0.88, 1.09)	0.747	<**0.001**	2103.63	0.0287
**F.** **Associated symptoms**								
Photophobia	All	1382/1582	Age	0.66 (0.54, 0.80)	<**0.001**	<**0.001**	1172.02	0.0540
			Duration	1.15 (0.96, 1.38)	0.123			
			Education years	0.92 (0.78, 1.07)	0.270			
			PRS	1.33 (1.13, 1.56)	**0.001**			
			Abortive use	2.15 (0.79, 5.84)	0.132			
			Preventive use	1.86 (1.36, 2.54)	<**0.001**			
			Anxiety	0.90 (0.65, 1.24)	0.530			
			−PRS	–	–	<**0.001**	1182.30	0.0437
Phonophobia	All	1200/1582	Age	0.80 (0.71, 0.90)	<**0.001**	<**0.001**	1698.44	0.0506
			Education years	0.92 (0.82, 1.05)	0.234			
			PRS	1.16 (1.02, 1.31)	**0.016**			
			Female sex	1.86 (1.33, 2.60)	<**0.001**			
			Abortive use	3.37 (1.32, 8.61)	**0.011**			
			Preventive use	1.88 (1.48, 2.41)	<**0.001**			
			Depression	1.28 (0.97, 1.69)	0.086			
			Vertigo	0.74 (0.53, 1.03)	0.076			
			−PRS	–	–	<0.001	1702.24	0.0473
	PRS deciles 1 and 10	247/319	Duration	0.82 (0.54, 0.96)	**0.024**	<**0.001**	333.28	0.1334
			Education years	0.85 (0.63, 1.15)	0.306			
			PRS decile 10	2.53 (1.36, 4.73)	**0.004**			
			Female sex	2.62 (1.22, 5.62)	**0.013**			
			Abortive use	10.6 (0.98, 113.5)	0.051			
			Preventive use	2.00 (1.09, 3.69)	**0.026**			
			Depression	2.08 (0.95, 4.55)	0.066			
			Stroke	0.12 (0.01, 1.86)	0.128			
			−PRS decile 10	–	–	**0.004**	340.12	0.1074
Nausea	All	1292/1582	Age	0.75 (0.63, 0.89)	**0.001**	<**0.001**	1471.81	0.0501
			Duration	1.18 (1.01, 1.39)	**0.043**			
			Education years	1.01 (0.88, 1.15)	0.910			
			PRS	1.15 (1.01, 1.32)	**0.039**			
			Female sex	1.92 (1.35, 2.73)	<**0.001**			
			Abortive use	2.14 (0.82, 5.54)	0.118			
			Preventive use	1.64 (1.25, 2.14)	<**0.001**			
			Stroke	0.44 (0.16, 1.17)	0.102			
			Hypertension	0.65 (0.46, 0.92)	**0.016**			
			−PRS	–	–	<**0.001**	1474.07	0.0472
	PRS deciles 1 and 10	267/319	Age	0.76 (0.54, 1.08)	0.132	**0.004**	281.90	0.1190
			Education years	0.94 (0.68, 1.30)	0.707			
			Abortive use	6.51 (0.85, 49.8)	0.071			
			Obesity	2.43 (0.78, 7.59)	0.127			
			Hypertension	0.41 (0.18, 0.91)	**0.029**			
			+PRS decile 10	1.51 (0.76, 2.99)	0.234	**0.004**	282.48	0.1240
Emesis	All	558/1582	Age	0.85 (0.76, 0.95)	**0.004**	**0.001**	2048.90	0.0189
			Education years	0.96 (0.87, 1.07)	0.496			
			PRS	1.12 (1.01, 1.25)	**0.033**			
			Abortive use	8.40 (1.11, 63.3)	**0.039**			
			Preventive use	1.27 (1.92, 1.57)	**0.033**			
			Asthma	1.22 (0.92, 1.63)	0.170			
			−PRS	–	–	**0.003**	2051.48	0.0167
	PRS deciles 1 and 10	122/319	Education years	0.80 (0.62, 1.03)	0.079	**0.009**	425.06	0.0692
			PRS decile 10	1.66 (1.01, 2.73)	**0.048**			
			Depression	2.09 (1.23, 3.58)	**0.007**			
			Asthma	2.44 (1.21, 4.92)	**0.013**			
			−PRS decile 10	–	–	**0.020**	427.00	0.0599
Neck stiffness	All	291/1582	Age	0.76 (0.63, 0.91)	**0.004**	<**0.001**	1458.37	0.0595
			Duration	1.34 (1.12, 1.61)	**0.001**			
			Education years	1.02 (0.90, 1.17)	0.712			
			Female sex	1.60 (0.99, 2.60)	0.057			
			Depression	1.41 (1.06, 1.87)	**0.018**			
			Fibromyalgia	1.88 (1.14, 3.09)	**0.013**			
			Obesity	1.36 (0.91, 2.02)	0.136			
			Vertigo	2.81 (2.03, 3.89)	<**0.001**			
			+PRS	1.09 (0.95, 1.25)	0.211	<**0.001**	1458.80	0.0606
	PRS deciles 1 and 10	64/319	Education years	1.03 (0.76, 1.37)	0.866	**0.007**	318.63	0.0850
			Vertigo	2.27 (1.14, 4.54)	**0.020**			
			+PRS decile 10	1.31 (0.71, 2.41)	0.391	**0.009**	319.89	0.0873
Osmophobia	All	1200/1582	Duration	1.19 (1.06, 1.34)	**0.004**	**0.001**	1875.14	0.0341
			Education years	0.93 (0.83, 1.04)	0.220			
			Female sex	2.05 (1.35, 3.11)	**0.001**			
			Preventive use	1.27 (1.00, 1.61)	0.050			
			Anxiety	1.21 (0.96, 1.53)	0.112			
			Fibromyalgia	1.49 (0.95, 2.34)	0.084			
			Vertigo	1.50 (1.10, 2.04)	**0.010**			
			Hypertension	0.65 (0.47, 0.89	**0.009**			
			+PRS	1.08 (0.96, 1.21)	0.199	<**0.001**	1875.49	0.0350
	PRS deciles 1 and 10	97/321	Education years	0.77 (0.59, 1.01)	0.059	**0.002**	388.39	0.0838
			PRS decile 10	1.84 (1.08, 3.14)	**0.025**			
			–PRS decile 10	–	–	**0.006**	391.52	0.0708
Allodynia	All	184/1582	Age	0.82 (0.65, 1.03)	0.089	<**0.001**	1078.06	0.0874
			Duration	1.28 (1.03, 1.60)	**0.028**			
			BMI	0.68 (0.55, 0.86)	**0.001**			
			Education years	1.06 (0.90, 1.26)	0.472			
			Female sex	5.27 (1.92, 14.5)	**0.001**			
			Anxiety	1.83 (1.32, 2.53)	<**0.001**			
			Asthma	0.65 (0.39, 1.07)	0.091			
			Obesity	2.43 (1.34, 4.39)	**0.003**			
			Vertigo	2.37 (1.62, 3.46)	<**0.001**			
			+PRS	1.06 (0.90, 1.25)	0.478	<**0.001**	1079.56	0.0878
	PRS deciles 1 and 10	39/319	BMI	0.62 (0.37, 1.02)	0.060	**0.002**	233.32	0.1504
			Education years	0.97 (0.67, 1.41)	0.872			
			Female sex	5.07 (0.64, 40.3)	0.124			
			Obesity	5.44 (1.53, 19.4)	**0.009**			
			Vertigo	4.01 (1.80, 8.90)	**0.001**			
			+PRS decile 10	1.59 (0.72, 3.51)	0.253	**0.002**	234.00	0.1560
Fatigue	All	527/1582	BMI	0.84 (0.74, 0.96)	**0.013**	<**0.001**	1979.65	0.0347
			Education years	1.01 (0.91, 1.13)	0.790			
			Female sex	1.34 (0.94, 1.92)	0.100			
			Anxiety	1.58 (1.26, 1.97)	<**0.001**			
			Fibromyalgia	1.54 (0.99, 2.41)	0.057			
			Obesity	1.47 (0.98, 2.21)	**0.066**			
			Vertigo	2.14 (1.59, 2.88)	<**0.001**			
			+PRS	1.05 (0.94, 1.18)	0.345	<**0.001**	1980.76	0.0351
	PRS deciles 1 and 10	103/319	BMI	0.71 (0.54, 0.94)	**0.015**	**0.002**	398.75	0.0961
			Education years	0.91 (0.70, 1.19)	0.502			
			PRS decile 10	1.90 (1.11, 3.24)	**0.020**			
			Preventive use	0.61 (0.36, 1.06)	0.080			
			Anxiety	1.68 (0.98, 2.89)	0.059			
			Fibromyalgia	2.92 (1.01, 8.48)	**0.049**			
			Asthma	2.31 (1.10, 4.88)	**0.028**			
			−PRS decile 10	–	–	**0.007**	402.31	0.0822
Visual changes	All	299/1582	Education years	0.85 (0.75, 0.97)	**0.014**	<**0.001**	1527.08	0.0227
			Anxiety	1.51 (1.16, 1.97)	**0.002**			
			Vertigo	1.71 (1.22, 2.39)	**0.002**			
			+PRS	1.02 (0.89, 1.16)	0.807	**0.002**	1529.02	0.0227
Dizziness	All	404/1582	Age	0.79 (0.70, 0.89)	<**0.001**	<**0.001**	1740.63	0.0518
			Education years	0.90 (0.80, 1.01)	0.076			
			Female sex	1.47 (0.99, 2.19)	0.056			
			Anxiety	1.46 (1.15, 1.87)	**0.002**			
			Fibromyalgia	1.47 (0.90, 2.39)	0.123			
			Asthma	1.30 (0.95, 1.78)	0.107			
			Vertigo	2.57 (1.88, 3.52)	<**0.001**			
			+PRS	1.01 (0.90, 1.14)	0.823	<**0.001**	1742.58	0.0518
	PRS deciles 1 and 10	83/319	Education years	0.83 (0.63, 1.10)	0.192	**0.039**	370.51	0.0635
			Asthma	3.15 (1.54, 6.46)	0.002			
			Vertigo	2.17 (1.13, 4.14)	**0.019**			
			+PRS decile 10	1.25 (0.72, 2.18)	0.427	**0.047**	371.88	0.0652

^a^ Logistic regression models (*symptom/characteristic ~ risk score + covariates + ten genetic PCs*) were fit using Akaike information criterion (AIC) backwards selection. Analyses used baseline data except for analyses of chronification and ED visits, which also included follow-up data. Continuous covariates included standardized age, disease duration, body mass index (BMI), number of years of education, and PGS004799 (PRS). Binomial covariates included sex, the presence of aura (not used in evaluation of migraine subtypes, aura, and types of aura), use of abortives and preventives, and migraine-related comorbidities present in at least 1% of patients: depression, anxiety, fibromyalgia, asthma, sleep apnea, obesity, vertigo, stroke, bipolar disorder, insomnia, and hypertension. If a model was significant, a sensitivity analysis evaluated the model using the first and tenth PRS decile as a binomial covariate (first decile = reference). The results of sensitivity analyses producing non-significant models are not reported. All models passed the RESET (Ramsey Regression Equation Specification Error Test), link test, and a Hosmer-Lemeshow goodness-of-fit test. The table reports data for the best fitting model: the number of patients with data on the retained covariates and among those, the number reporting the symptom, model *p*, covariates retained with their odds ratios and 95% confidence intervals (CI), and model *p*, AIC and McFadden’s pseudo-*R*^2^. For comparison, the table also reports, in the line below the dashed line, the model *p*, AIC and pseudo-*R*^2^ for models with (or without) the PRS and its β and CI, if it was excluded (or retained) during model fitting. *p* < 0.05 in bold.

**Table 3 jcm-14-00536-t003:** Association of migraine risk scores with baseline assessments of migraine-related disability and migraine characteristics ^a^.

**A. Negative binomial regression models**								
**Measure**	**Group**	**N**	**Covariate**	**Coefficient** **(95% CI)**	** *p* **	**Model**		
						** *p* **	**AIC**	**McFadden’s Pseudo *R*^2^**
MIDAS Score	All	1499	Age	−0.15 (−0.21, −0.09)	<**0.001**	<**0.001**	13,545.85	0.0117
			BMI	0.06 (0.01, 0.12)	**0.032**			
			Education years	−0.10 (−0.16, −0.05)	<**0.001**			
			Preventive use	0.39 (0.28, 0.51)	<**0.001**			
			Anxiety	0.21 (0.10, 0.33)	<**0.001**			
			Fibromyalgia	0.55 (0.31, 0.79)	<**0.001**			
			+PRS	−0.02 (−0.07, 0.04)	0.547	<**0.001**	13,547.49	0.0117
	PRS deciles 1 and 10	297	Age	−0.24 (−0.36, −0.12)	<**0.001**	<**0.001**	2681.98	0.0257
			Education years	−0.18 (−0.29, −0.07)	**0.002**			
			Female sex	0.39 (0.03, 0.74)	**0.032**			
			Abortive use	2.42 (1.39, 3.45)	<**0.001**			
			Preventive use	0.19 (−0.06, 0.44)	0.133			
			Fibromyalgia	0.50 (−0.02, 1.01)	0.060			
			+PRS decile 10	0.05 (−0.19, 0.30)	0.670	<**0.001**	2683.80	0.0257
ISI Score	All	1547	Duration	0.06 (0.02, 0.10)	**0.006**	**0.018**	9258.31	0.0031
			Education years	−0.04 (−0.09, −0.0001)	0.050			
			PRS	−0.04 (−0.09, −0.002)	**0.038**			
			Anxiety	0.10 (0.01, 0.19)	**0.023**			
			Vertigo	0.09 (−0.03, 0.21)	0.139			
			−PRS	−	−	**0.043**	9260.60	0.0026
**B. Linear regression models**								
**Measure**	**Group**	**N**	**Covariate**	**Coefficient** **(95% CI)**	** *p* **	**Model**		
						** *p* **	**AIC**	** *R* ^2^ **
ln(MIDAS-A)	All	1473	Age	−0.13 (−0.20, −0.06)	<**0.001**	<**0.001**	4169.69	0.1001
			Duration	0.08 (0.01, 0.15)	**0.017**			
			Education years	−0.04 (−0.09, 0.02)	0.181			
			Aura	−0.18 (−0.28, −0.07)	**0.001**			
			Abortive use	0.84 (0.39, 1.30)	<**0.001**			
			Preventive use	0.42 (0.32, 0.53)	<**0.001**			
			Fibromyalgia	0.38 (0.16, 0.61)	**0.001**			
			Obesity	0.20 (0.03, 0.36)	**0.020**			
			+PRS	0.02 (−0.03, 0.07)	0.179	<**0.001**	4171.21	0.0993
	PRS deciles 1 and 10	288	Education years	−0.09 (−0.20, 0.01)	0.076	<**0.001**	744.85	0.2924
			PRS decile 10	0.17 (−0.05, 0.38)	0.129			
			Aura	−0.29 (−0.50, −0.08)	**0.005**			
			Abortive use	1.68 (0.90, 2.47)	<**0.001**			
			Preventive use	0.52 (0.31, 0.74)	<**0.001**			
			Depression	−0.29 (−0.54, −0.04)	**0.023**			
			Anxiety	0.17 (−0.05, 0.39)	0.136			
			Stroke	−0.95 (−2.17, 0.27)	0.128			
			Asthma	0.38 (0.08, 0.69)	**0.014**			
			Obesity	0.31 (0.01, 0.62)	**0.045**			
			−PRS decile 10	–	–	<**0.001**	745.34	0.2862
MSQ score	All	1547	Age	2.13 (1.37, 2.89)	<**0.001**	<**0.001**	12,696.97	0.1412
			BMI	−1.01 (−1.75, −0.28)	**0.007**			
			Education years	1.75 (1.00, 2.50)	<**0.001**			
			Female sex	−3.51 (−5.76, −1.25)	**0.002**			
			Abortive use	−5.86 (−12.3, 0.61)	0.076			
			Preventive use	−5.45 (−6.98, −3.92)	<**0.001**			
			Depression	−3.34 (−5.10, −1.58)	<**0.001**			
			Anxiety	−2.10 (−3.77, −0.43)	**0.014**			
			Fibromyalgia	−8.60 (−11.9, −5.29)	<**0.001**			
			+PRS	−0.18 (−0.57, 0.93)	0.633	<**0.001**	12,698.74	0.1411
	PRS deciles 1 and 10	310	Age	3.77 (2.13, 5.40)	<**0.001**	<**0.001**	2546.44	0.2450
			BMI	−1.27 (−2.85, 0.31)	0.114			
			Education years	2.98 (1.32, 4.64)	<**0.001**			
			Female sex	−8.21 (−13.0, −3.39)	**0.001**			
			Abortive use	−15.5 (−28.5, −2.50)	**0.020**			
			Preventive use	−4.39 (−7.89, −0.90)	**0.014**			
			Fibromyalgia	−7.42 (−14.6, −0.25)	**0.043**			
			+PRS decile 10	−0.95 (−4.39, 2.50)	0.590	<**0.001**	2548.13	0.2458
**C. Ordinal logistic regression models**								
**Measure**	**Group**	**N**	**Covariate**	**Odds Ratio** **(95% CI)**	** *p* **	**Model**		
						** *p* **	**AIC**	**McFadden’s Pseudo *R*^2^**
Frequency	All	1576	Duration	1.10 (1.01, 1.21)	**0.035**	<**0.001**	5686.03	0.0232
			Education years	0.82 (0.75, 0.90)	<**0.001**			
			Aura	0.73 (0.61, 0.87)	**0.001**			
			Abortive use	3.41 (1.39, 8.33)	**0.007**			
			Preventive use	1.91 (1.59, 2.29)	<**0.001**			
			Anxiety	1.39 (1.15, 1.68)	**0.001**			
			Obesity	1.63 (1.22, 2.17)	**0.001**			
			+PRS	0.94 (0.86, 1.03)	0.187	<**0.001**	5686.20	0.0235
	PRS deciles 1 and 10	319	Age	1.28 (0.99, 1.66)	0.062	<**0.001**	1141.10	0.0364
			Duration	0.80 (0.62, 1.03)	0.083			
			Education years	0.88 (0.72, 1.07)	0.191			
			Abortive use	17.9 (2.63, 121.9)	**0.003**			
			Preventive use	2.84 (1.85, 4.37)	<**0.001**			
			Depression	0.65 (0.40, 1.06)	0.087			
			Anxiety	1.87 (1.18, 2.94)	**0.007**			
			+PRS decile 10	1.10 (0.73, 1.64)	0.654	<**0.001**	1142.90	0.0366
MIDAS-B	All	1523	Duration	0.90 (0.82, 0.99)	**0.024**	<**0.001**	6304.52	0.0182
			BMI	1.12 (1.02, 1.22)	**0.014**			
			Education years	0.76 (0.69, 0.84)	<**0.001**			
			Female sex	1.75 (1.32, 2.32)	<**0.001**			
			Aura	0.79 (0.65, 0.94)	**0.010**			
			Abortive use	4.55 (1.95, 10.6)	<**0.001**			
			Preventive use	1.19 (0.99, 1.43)	0.070			
			Fibromyalgia	1.59 (1.07, 2.38)	**0.023**			
			+PRS	1.01 (0.92, 1.10)	0.883	<**0.001**	6306.50	0.0182
	PRS deciles 1 and 10	304	Age	0.85 (0.69, 1.04)	0.127	<**0.001**	1277.56	0.0373
			Education years	0.79 (0.64, 0.97)	**0.026**			
			Female sex	2.60 (1.38, 4.91)	**0.003**			
			Aura	0.72 (0.47, 1.10)	0.124			
			Abortive use	6.12 (1.26, 29.7)	**0.024**			
			Preventive use	0.69 (0.45, 1.07)	0.100			
			Fibromyalgia	2.49 (0.95, 6.51)	0.063			
			Obesity	1.65 (0.01, 2.97)	0.102			
			+PRS decile 10	0.88 (0.69, 1.05)	0.566	<**0.001**	1279.23	0.0376
Symptom severity	All	1557	Duration	0.80 (0.73, 0.88)	<**0.001**	<**0.001**	6605.99	0.0179
			BMI	1.10 (1.01, 1.21)	**0.029**			
			Education years	0.78 (0.72, 0.86)	<**0.001**			
			PRS	1.09 (1.00, 1.20)	0.052			
			Female sex	0.45 (1.11, 1.90)	**0.006**			
			Aura	0.82 (0.69, 0.99)	**0.036**			
			Abortive use	4.90 (2.07, 11.59)	<**0.001**			
			−PRS	–	–	<**0.001**	6607.78	0.0174
	PRS deciles 1 and 10	320	Duration	0.79 (0.65, 0.96)	**0.016**	<**0.001**	1357.80	0.0450
			Education years	0.80 (0.65, 0.97)	**0.023**			
			Female sex	2.52 (1.38, 4.58)	**0.003**			
			Aura	0.72 (0.48, 1.07)	0.103			
			Abortive use	25.1 (4.71, 134.0)	<**0.001**			
			+PRS decile 10	1.34 (0.89, 2.04)	0.164	<**0.001**	1357.86	0.0436
Time-to-peak intensity	All	1125	Education years	1.18 (1.06, 1.33)	**0.002**	**0.001**	2989.25	0.0120
			Aura	0.70 (0.56, 0.88)	**0.002**			
			Preventive use	1.29 (1.03, 1.61)	**0.025**			
			Depression	0.80 (0.62, 1.02)	0.067			
			+PRS	1.08 (0.97, 1.20)	0.180	**0.001**	2989.45	0.0126
	PRS deciles 1 and 10	238	Age	0.80 (0.62, 1.02)	0.071	**0.034**	636.15	0.0423
			Education years	1.09 (0.87, 1.38)	0.499			
			PRS decile 10	1.51 (0.91, 2.49)	0.111			
			Depression	0.58 (0.32, 1.02)	0.058			
			Asthma	2.30 (1.06, 5.01)	**0.035**			
			−PRS decile 10	–	–	**0.038**	636.71	0.0475
Average duration	All	1569	Education years	0.97 (0.88, 1.06)	0.480	**0.001**	4209.90	0.0196
			Female sex	1.77 (1.33, 2.35)	<**0.001**			
			Aura	0.75 (0.62, 0.91)	**0.003**			
			Abortive use	3.39 (1.41, 8.15)	**0.006**			
			Preventive use	1.50 (1.25, 1.82)	<**0.001**			
			Fibromyalgia	1.62 (1.09, 2.41)	**0.018**			
			Asthma	1.43 (1.10, 1.85)	**0.008**			
			+PRS	1.02 (0.93, 1.13)	0.622	<**0.001**	4210.78	0.0197
	PRS deciles 1 and 10	315	BMI	1.20 (0.97, 1.47)	0.093	**0.001**	870.75	0.0464
			Education years	0.92 (0.75, 1.13)	0.426			
			Female sex	2.85 (1.52, 5.36)	**0.001**			
			Abortive use	6.33 (1.33, 30.2)	**0.020**			
			Preventive use	1.63 (1.04, 2.54)	**0.033**			
			Depression	0.49 (0.29, 0.81)	**0.005**			
			Asthma	1.74 (0.92, 3.31)	0.091			
			+PRS decile 10	1.18 (0.76, 1.82)	0.466	**0.002**	872.21	0.0470
Average aura duration	All	585	Education years	1.12 (0.95, 1.32)	0.177	**0.004**	1246.92	0.0247
			Stroke	5.18 (1.35, 19.83)	**0.016**			
			Asthma	0.49 (0.31, 0.79)	**0.003**			
			+PRS	0.99 (0.84, 1.17)	0.952	**0.006**	1248.91	0.0247

^a^ Regression models (*measure ~ risk score + covariates + ten genetic PCs*) were fit using Akaike information criterion (AIC) backwards selection. In all models, continuous covariates were standardized and included initial-visit age, disease duration, body mass index (BMI), number of years of education, and PGS004799 (PRS). Binomial covariates included sex, the presence of aura, use of abortives and preventives, and migraine-related comorbidities present in at least 1% of patients: depression, anxiety, fibromyalgia, asthma, sleep apnea, obesity, vertigo, and hypertension. The type of regression used in a model was selected based on the distribution of the dependent variable (see Figure 3 and Figure 4). Negative binomial regression was used to evaluate associations with count variables showing non-normal distributions (CES-D, ISI, and MIDAS). Linear regression was used to evaluate associations with MSQ scores and ln(MIDAS-A). Ordinal logistic regression was used to evaluate associations with dependent variables having natural ordering: frequency, MIDAS-B and symptom severity, average duration, time-to-peak intensity, and average aura duration. For these, odds ratios are reported for the first equation (level one) only. If a model was significant, a sensitivity analysis evaluated whether the dependent variable differed between tenth and first decile scores (first decile = reference). Non-significant models and models failing goodness-of-fit tests (see text) are not reported. The table reports data for the best fitting model: the number of patients with data for the measure and the retained covariates that were assessed in the model, covariates retained and their coefficients or odds ratios and 95% confidence interval (CI), and model *p*, AIC and either *R*^2^ or McFadden’s pseudo-*R*^2^. For comparison, the table also reports, in the line below the dashed line, the model *p*, AIC and either *R*^2^ or pseudo-*R*^2^ for models with (or without) the PRS and its odds ratio or β and CI, if it was excluded (or retained) during model fitting. *p* < 0.05 in bold.

**Table 4 jcm-14-00536-t004:** Associations of migraine PRS with changes at first annual follow-up ^a^.

Measure	Baseline and Follow-Up Values					Associations With Change at Follow-Up							
	N	Median (Range)		Matched-Pairs Signed-Rank Test									
		Baseline	Follow Up	Z	*p*	N	Covariate		β (95% CI)	*p*	Negative Binomial Regression Model		
											*p*	AIC	McFadden’s Pseudo *R*^2^
ISI score	674	8 (0–28)	7 (0–28)	2.48	**0.013**	669	Baseline ISI score		0.03 (0.02, 0.04)	<**0.001**	<**0.001**	4196.55	0.0103
							Duration		0.11 (0.05, 0.17)	**0.001**			
							Education years		0.03 (−0.03, 0.09)	0.342			
							+PRS		0.03 (−0.04, 0.09)	0.390	<**0.001**	4197.81	0.0105
MIDAS score	613	23 (0–420)	16 (1–450)	5.62	**1.5 × 10^−8^**		No models passed a link test						
CES-D score	687	9 (0–53)	8 (0–58)	0.38	0.701		Not evaluated (no change at follow-up)						
GAD-7 score	686	3 (0–21)	3 (0–21)	−0.061	0.951								
						**N**	**Covariate**		**β (95% CI)**	** *p* **	**Linear Regression Model**		
											** *p* **	**AIC**	** *R* ^2^ **
MSQ score	734	58 (4–84)	67 (14–84)	−12.31	**8.0 × 10^−35^**	727	Baseline MSQ				<**0.001**	5830.92	0.3135
							score		0.50 (0.43, 0.56)	<**0.001**			
							Education years		0.98 (−0.02, 1.98)	0.055			
							Female sex		−3.64 (−6.99, −0.28)	**0.034**			
							Preventive use		−3.07 (−5.13, −1.02)	**0.003**			
							Abortive use		−8.26 (−19.1, 2.62)	0.137			
							Sleep apnea		−3.23 (−6.40, −0.05)	**0.046**			
							Hypertension		1.98 (−0.61, 4.57)	0.134			
							+PRS		−0.36 (−1.31, 0.59)	0.456	<**0.001**	5832.36	0.3140
ln(MIDAS-A)	677	3.04 (0–4.50)	2.71 (0–4.5)	9.13	**1.3 × 10^−20^**	671	Baseline				<**0.001**	1830.90	0.2524
							ln(MIDAS-A)		0.44 (0.37, 0.52)	<**0.001**			
							Age		−0.06 (−0.13, 0.02)	0.121			
							Education years		0.01 (−0.07, 0.09)	0.797			
							PRS		0.06 (−0.02, 0.13)	0.123			
							Female sex		0.19 (−0.05, 0.43)	0.121			
							Preventive use		0.29 (0.14, 0.45)	<**0.001**			
							−PRS		−	−	<**0.001**	1831.34	0.2497
						**N**	**Covariate**		**Odds Ratio (95% CI)**	** *p* **	**Ordinal Logistic Regression Model**		
											** *p* **	**AIC**	**McFadden’s Pseudo *R*^2^**
Frequency	772	4 (1–7)	3 (1–7)	11.72	**1.2 × 10^−33^**	765	Baseline	2	5.23 (2.39, 11.5)	<**0.001**	<**0.001**	2653.91	0.0872
							Frequency	3	11.7 (5.13, 26.9)	<**0.001**			
							(relative to	4	14.9 (6.8, 32.6)	<**0.001**			
							group 1)	5	28.1 (12.8, 61.8)	<**0.001**			
							6		37.5 (16.8, 83.7)	<**0.001**			
							7		114.8 (44.9, 293.1)	<**0.001**			
							Age		0.83 (0.70, 0.98)	**0.029**			
							Duration		1.16 (0.98, 1.37)	0.089			
							Education years		1.02 (0.89, 1.16)	0.825			
							Preventive use		1.67 (1.26, 2.21)	<**0.001**			
							Obesity		1.59 (1.04, 2.44)	**0.031**			
							+PRS		0.99 (0.88, 1.13)	0.939	<**0.001**	2655.90	0.0872
MIDAS-B	722	7 (0–10)	6 (0–10)	9.11	**1.7 × 10^−20^**	715	Baseline	1	2.50 (1.52, 285.3)	0.473	<**0.001**	2833.05	0.0831
							Severity	2	2.13 (0.96, 120.3)	0.522			
							(relative to	3	7.27 (2.4, 303.3)	0.071			
							group 0)	4	11.7 (7.98, 996.7)	**0.025**			
							5		13.7 (10.1, 1188)	**0.015**			
							6		35.3 (22.0, 2575)	**0.001**			
							7		51.5 (34.9, 4057)	<**0.001**			
							8		97.3 (103, 12,140)	<**0.001**			
							9		171.0 (159, 19,841)	<**0.001**			
							10		339.4 (359, 46,473)	<**0.001**			
							Age		0.80 (0.73, 0.95)	**0.017**			
							Duration		1.28 (0.83, 1.08)	**0.004**			
							Education years		0.90 (0.59, 0.99)	0.129			
							Female sex		1.58 (1.13, 2.53)	**0.048**			
							Aura		0.75	**0.037**			
							Sleep apnea		1.42	0.114			
							Hypertension		0.71	0.061			
							+PRS		0.99 (0.87, 1.12)	0.854	<**0.001**	2835.01	0.0831
Severity	769	7 (1–10)	6 (0–10)	9.17	**8.0 × 10^−22^**	762	Baseline	1	20.8 (1.52, 285.3)	**0.023**	<**0.001**	2939.05	0.1193
							Severity	2	10.7 (0.96, 120.3)	0.054			
							(relative to	3	27.0 (2.4, 303.3)	**0.008**			
							group 0)	4	89.2 (7.98, 996.7)	<**0.001**			
							5		109.8 (10.1, 1188)	<**0.001**			
							6		238.2 (22.0, 2575)	<**0.001**			
							7		376.4 (34.9, 4057)	<**0.001**			
							8		1117 (103, 12,140)	<**0.001**			
							9		1775 (159, 19,841)	<**0.001**			
							10		4084 (359, 46,473)	<**0.001**			
							Age		0.83 (0.73, 0.95)	**0.006**			
							Education years		0.95 (0.83, 1.08)	0.441			
							Aura		0.77 (0.59, 0.99)	**0.044**			
							Sleep apnea		1.69 (1.13, 2.53)	**0.011**			
							+PRS		0.99 (0.88, 1.13)	0.933	<**0.001**	2941.04	0.1193
Average duration	757	5 (3–7)	5 (1–7)	5.77	**5.3 × 10^−9^**	No models passed a Lipsitz goodness-of-fit test							
Time-to-peak intensity	529	3 (1–5)	3 (1–5)	0.41	0.690	Not evaluated (no change at follow-up)							
Aura duration	219	2 (1–5)	2 (1–5)	−0.21	0.950								

^a^ The right columns report the number of patients with data on a measure at both baseline and follow-up, the median (range) values at these encounters, and the results of a Wilcoxon matched-pairs signed-rank test. Compare to the graphs in the left column of Figure 3 and Figure 4. If the paired rank-sum test demonstrated that measures differed at follow-up, regression analyses evaluated whether the PRS was associated with follow-up. The left columns report the results of fitting follow-up values to regression models, whose type was selected based on the distribution of the follow-up measure (see Methods, Figure 3 and Figure 4). Negative binomial regression was used for ISI, MIDAS, CES-D, and GAD7 scores, linear regression was used for ln(MIDAS-A) and MSQ scores, and ordinal logistic regression was used for frequency, MIDAS-B, symptom severity, time-to-peak intensity, average duration, and average duration of aura. Regression models (*measure at follow-up ~ measure at baseline + risk score + covariates + ten genetic PCs*) were fit using Akaike information criterion (AIC) backwards selection. Continuous covariates were standardized and included initial-visit age, disease duration, body mass index (BMI), number of years of education, and PGS004799 (PRS). Binomial covariates included sex, the presence of aura (except for average aura duration), use of abortives and preventives, and migraine-related comorbidities present in at least 1% of patients: depression, anxiety, fibromyalgia, asthma, sleep apnea, obesity, vertigo, stroke, bipolar disorder, insomnia, and hypertension. All models evaluated for follow-up MIDAS score failed a link test, and all models evaluated for follow-up average duration failed a Lipsitz goodness-of-fit test, so none are reported. All reported linear regression models passed a link test and a Ramsey Regression Equation Specification Error Test (RESET) and had near-normally distributed residuals. Reported ordinal logistic regression models passed a link test and a Lipsitz goodness-of-fit test. For each fitted model, the table reports the number of patients analyzed, model *p*, AIC and *R*^2^ or pseudo-*R*^2^, the covariates retained, and their β coefficients or odds ratios (first equation only) and 95% confidence interval (CI). The number of patients used in the models is slightly lower than the number having data on the measure at baseline and follow-up because patients with incomplete covariate data were excluded from the analysis. For comparison, the table also reports, in the line below the dashed line, the model *p*, AIC and *R*^2^ or pseudo-*R*^2^ for models with (or without) the PRS, and its β and CI, if it was excluded (or retained) during model fitting. *p* < 0.05 in bold font.

## Data Availability

Summary statistics are available from the corresponding author upon reasonable request, following institutional approval.

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
