# Peer review of "Migraine Genetic Susceptibility Does Not Strongly Influence Migraine Characteristics and Outcomes in a Treated, Real-World, Community Cohort"

_jcm, 2025, doi:10.3390/jcm14020536_

Round 1

Reviewer 1 Report

Comments and Suggestions for Authors

Chase and coll wrote a very well performed and written study on a very interesting topic. This article explores an uncharted aspect of an important topic: the role of genetic background in the clinical manifestations of migraine in a specific cohort of treated patients.

Despite several methodological limitations acknowledged by the authors themselves, they do so in a precise and detailed manner, with many well-described results.

This reviewer just finds some flaws in the Discussion section, It would be better to specify the results when discussing them, perhaps by putting them in brackets, to make the data easier for the reader to understand. For instance, in lines 414-415: "We found that the PRS was associated with more common migraine characteristics and symptoms", which characteristics and symptoms?.

I would also appreciate it if you could explain the possible spillover of your results into clinical practice in the Discussion section. Such a speculation has indeed been addressed, but it is more about the type of study in general, not about your specific results.

Author Response

Comment 1: Chase and coll wrote a very well performed and written study on a very interesting topic. This article explores an uncharted aspect of an important topic: the role of genetic background in the clinical manifestations of migraine in a specific cohort of treated patients.

Despite several methodological limitations acknowledged by the authors themselves, they do so in a precise and detailed manner, with many well-described results.

Response 1: Thank you for the comments. We appreciate them.

Comment 2: This reviewer just finds some flaws in the Discussion section, It would be better to specify the results when discussing them, perhaps by putting them in brackets, to make the data easier for the reader to understand. For instance, in lines 414-415: "We found that the PRS was associated with more common migraine characteristics and symptoms", which characteristics and symptoms?.

Response 2:  Thank you for this suggestion.  We have addressed this on lines 419-424.  For clarity, we have inserted an additional sentence instead of adding them parenthetically: "More specifically, in this population, higher PRS was associated with photophobia and stabbing pain, but the PRS was not associated with other symptoms of migraine attacks, objective measures of migraine disability, frequency, severity, average duration, time-to-peak intensity of migraine attacks, chronification, emergency-department visits, triptan responsiveness, or at follow-up, improvement in MIDAS or MSQ scores, migraine frequency, severity or average duration."

Comment 3: I would also appreciate it if you could explain the possible spillover of your results into clinical practice in the Discussion section. Such a speculation has indeed been addressed, but it is more about the type of study in general, not about your specific results.

Response 3:  Thank you also for this suggestion.  We have addressed this on lines 455-457: "The findings of our study supports the view that clinicians should counsel patients living with migraine that genetic susceptibility to migraine does not determine disease course."

Reviewer 2 Report

Comments and Suggestions for Authors

The Authors presented a paper:" Migraine genetic susceptibility does not strongly influence migraine characteristics and outcomes in a treated, real-world, community cohort" very interesting and well done.

The abstract reflects the content of the work. The research is well designed, the methodology is adequate, the results are clearly presented. Testing the hypothesis that genetic susceptibility to migraine is associated with clinical characteristics or outcomes in treated patients is important for anyone who treating migraine patients. The conclusions reached by the research are useful for both researchers and doctors in clinical practice. Limitations of the study are also mentioned. References are current and adequate.

Author Response

Comment 1: 

The Authors presented a paper:" Migraine genetic susceptibility does not strongly influence migraine characteristics and outcomes in a treated, real-world, community cohort" very interesting and well done.

The abstract reflects the content of the work. The research is well designed, the methodology is adequate, the results are clearly presented. Testing the hypothesis that genetic susceptibility to migraine is associated with clinical characteristics or outcomes in treated patients is important for anyone who treating migraine patients. The conclusions reached by the research are useful for both researchers and doctors in clinical practice. Limitations of the study are also mentioned. References are current and adequate.

Response 1:  Thank you for the comment.  We appreciate them.

Reviewer 3 Report

Comments and Suggestions for Authors

Please excuse grammatical and spelling errors as a dictation software useD.

A good review article.

this was a reasonable review article. It’s difficult to state the genetics involve in migraine headaches because there aren’t a lot of genes that was discovered for migraines.

Consider, including CACNA1 genesThat’s not the cost for hemiplegic Migraines and spino cerebellar ataxia.

Meta-analysis curves, can be helpful in showing the relationships are various studies, the genetic predisposition.

consider adding more definition to polygenetic risk factor to readers understanding that there are no specific genes found for migraines.

in the conclusion place limitations of studies include lack of genetic studies, specifically studying migraines.

in terms of environmental factors, please date certain factors that may influence migraine such as dehydration, extreme heat, and extreme cold, and also high altitude.

discuss also psychiatric disease diseases that may influence headaches.

Generally a good paper with a few suggestions.

Author Response

Comments 1: Please excuse grammatical and spelling errors as a dictation software useD.

A good review article.  this was a reasonable review article.

It’s difficult to state the genetics involve in migraine headaches because there aren’t a lot of genes that was discovered for migraines.

Consider, including CACNA1 genesThat’s not the cost for hemiplegic Migraines and spino cerebellar ataxia.

Meta-analysis curves, can be helpful in showing the relationships are various studies, the genetic predisposition.

consider adding more definition to polygenetic risk factor to readers understanding that there are no specific genes found for migraines.

Response 1:  Thank you for these comments.   Our article presents a primary analysis of data that we gathered on the relationship of a polygenic risk score to migraine phenotypes found in patients from a community cohort who are living with migraine and being treated by neurologists in a headache specialty clinic.  While we review some pertinent literature in the Introduction and Discussion, our manuscript is neither a review article or a meta-analysis.  So, we can not include "meta-analysis curves" (we think the reviewer means "forest plots"?) in our manuscript.   This said, we concur that there are no single gene mutations that cause migraine, except for familial hemiplegic migraine where migraine with aura is accompanied by reversible motor symptoms and, at a later age, ataxia symptoms.  To add this context to our use of a polygenic risk score , we have added text on lines 54-57: "With the exception of familial hemiplegic migraine, where the inheritance of mutations at the CACNA1A, ATTP1A2 or SCN1A genes leads to a phenotype that includes reversible motor weakness and migraine, migraine has not been associated with single gene mutations." 

Comment 2: in the conclusion place limitations of studies include lack of genetic studies, specifically studying migraines. 

Response 2:  We haven't made changes in response to this comment, we responded considering that it may be referring to three different suggestions.  (i) In the Limitations section, we address limitations of our study, but not other studies, since this is not a review article or meta-analysis.  (ii) We describe in the Introduction that migraine is considered a complex genetic disorder, and this is supported by a great deal of (uncited) literature.  (iii) On lines 458-464, we suggest prospective, genetic-framed studies that would clarify the relationship between genetic susceptibility and treatment, genetic contributions to treatment response, and the relationship between genetic factors, environmental factors, and treatment outcomes.  

Comments 3: in terms of environmental factors, please date certain factors that may influence migraine such as dehydration, extreme heat, and extreme cold, and also high altitude.

Response 3:  Thank you for the comment.  Our study did not evaluate the contribution of environmental factors (such as dehydration, extreme heat or cold, or high altitude to migraine symptoms) relative to genetic risk.  We agree that these, as well as other, environmental factors can influence migraine symptoms.  Since our manuscript is not a review and we did not assess the relationship between environmental factors and migraine symptoms or outcomes in our patient population, we don't feel it is appropriate to discuss specific environmental factors.  We prefer to keep our statements about the contribution of such factors to be very general, as "environmental factors." However, we fully agree that these and other environmental factors can be important, and this is why we propose prospective studies to understand the relationship between them, genetic factors and treatment outcomes (line 462-463).

Comment 4: discuss also psychiatric disease diseases that may influence headaches.

Response 4: Thank you for the comment.  As this is not a review, we did not specifically discuss the factors that can influence migraine symptoms and frequency.  Since we were interested in the relationship of genetic susceptibility for migraine to migraine symptoms and outcomes, however, we did include such factors, which have been reported in other reports, as covariates in our analyses.  In particular we included psychiatric-disease comorbidities as covariates in our analyses if they were present in >1% of our patient population.  This is described in the manuscript on lines 140-141, 290-294, and in Figure 2J.  More specifically, we included both depression and bipolar disorder as covariates in our analyses. 

Comment 5:  Generally a good paper with a few suggestions.

Response 5:  Thank you for the comment.  We appreciate it.